# TAKEUCHI'S INFORMATION CRITERIA AS GENERALIZATION MEASURES FOR DNNS CLOSE TO NTK REGIME

## ABSTRACT

Generalization measures are intensively studied in the machine learning community for better modeling generalization gaps. However, establishing a reliable generalization measure for statistical singular models such as deep neural networks (DNNs) is challenging due to the complex nature of the singular models. We focus on a classical measure called Takeuchi's Information Criteria (TIC) to investigate allowed conditions in which the criteria can well explain generalization gaps caused by DNNs. In fact, the theory indicates the applicability of TIC near the neural tangent kernel (NTK) regime. Experimentally, we trained more than 5,000 DNN models with 12 DNN architectures including large models (e.g., VGG16) and 4 datasets, and estimated corresponding TICs in order to comprehensively study the relationship between the generalization gap and the TIC estimates. We examine several approximation methods to estimate TIC with feasible computational load and investigate the accuracy trade-off. Experimental results indicate that estimated TIC well correlates generalization gaps under the conditions that are close to NTK regime. Outside the NTK regime, such correlation disappears, shown theoretically and empirically. We further demonstrate that TIC can yield better trial pruning ability for hyperparameter optimization over existing methods.

## 1 INTRODUCTION

Deep neural networks (DNNs) have been exhibiting great generalization abilities in many applications, but the mechanism of the generalization has not been fully understood yet (Neyshabur et al., 2014; Zhang et al., 2016; Recht et al., 2019). Establishing a reliable generalization measure is an important research topic for generating a good model from limited data resources, including an application of hyperparameter search. Many attempts (Arora et al., 2018; Wei & Ma, 2019; Neyshabur et al., 2018) have been taken to better understand the generalization phenomenon in deep learning models from theoretical points of view. From empirical points of view, there have been intensive studies (Keskar et al., 2016; Liang et al., 2019; Bartlett et al., 2017) in search of learning conditions that likely yield high model performance.

Work by (Jiang et al., 2019) indicated that a measure that includes both hessian $H(\theta)$[1] and covariance $C(\theta)$ defined from the loss and the network parameters $\theta$ near a local minimum may potentially show good correlation with generalization performance. Another study indicated that use of only a single measure, either Hessian $H(\theta)$ or covariance $C(\theta)$, fails to capture the generalization performance (Novak et al., 2018a).

The generalization gap inherently stems from a discrepancy between the empirical and the true data distribution. A minimizer of the empirical loss will be affected by the noise due to a finite number of samples and by the form of the loss landscape near the minimum. The former can be characterized as noise ($C(\theta)$) and the latter as curvature ($H(\theta)$).

Taking these findings into account, we sought to model the generalization gap, then found that a classical information criterion called Takeuchi's Information Criteria (TIC) (Takeuchi, 1976) ex-

---

[1] $H(\theta)$ and $C(\theta)$ are defined in equation 2

presses generalization gap in the neural tangent kernel (NTK) regime. TIC has the following form

$$\underbrace{\text{TIC}(\boldsymbol{\theta})}_{\text{Information Criteria}} = -\underbrace{\mathbb{E}_{\hat{p}}[l(y, f(x, \boldsymbol{\theta}))]}_{\text{Mean Empirical Error}} + \underbrace{\text{Tr}\left(\boldsymbol{H}(\boldsymbol{\theta})^{-1}\boldsymbol{C}(\boldsymbol{\theta})\right)}_{\text{Estimated Bias Term}}, \tag{1}$$

where $f$ is a smooth function over $\boldsymbol{\theta} \in \mathbb{R}^d$ with input $x$ and target $y$, and $l$ is the negative log-likelihood, also denoted as loss function. The first term on the right-hand side is the log-likelihood, which takes the expectation over an empirical data distribution $(x_i, y_i) \sim \hat{p}$. For the later discussion, we use $\hat{\boldsymbol{\theta}}$ minimizes the empirical loss; *i.e.*, $\hat{\boldsymbol{\theta}} = \arg\min_{\theta}\mathbb{E}_{\hat{p}}[l(y, f(x, \boldsymbol{\theta}))]$, and $\boldsymbol{\theta}^*$ as the parameters that maximizes the likelihood with respect to the true data distribution $(x, y) \sim p$; *i.e.*, $\boldsymbol{\theta}^* = \arg\min_{\theta}\mathbb{E}_p[l(y, f(x, \boldsymbol{\theta}))]$.

For a DNN of practical size, exact computations of the matrices $\boldsymbol{H}(\boldsymbol{\theta})$, $\boldsymbol{C}(\boldsymbol{\theta})$ are nearly infeasible due to large dimensionality. To make the computation feasible, we adopted a strategy of using shared components of the matrix to estimate TIC with fewer computations, based on a relationship among matrices such as $\boldsymbol{H}(\boldsymbol{\theta})$, $\boldsymbol{C}(\boldsymbol{\theta})$, and Fisher Information Matrix $\boldsymbol{F}(\boldsymbol{\theta})$. To further reduce the computational costs for the bias term, we examined methods using approximations and lower bounds so that TIC estimations for DNNs of practical sizes are feasible.

In this work, we make the following contributions:

- We provide empirical and theoretical evidence that TIC is highly correlated with generalization gap of DNNs that are close to NTK regime, despite the fact that TIC is not originally designed for singular model such as general DNNs.
- We conduct *comprehensive experiments in which more than 5,000 models, including ones close to NTK regime, with totally 12 DNN architectures, 4 datasets and 15 training settings are trained, and corresponding TICs are estimated with approximation techniques*, to clarify conditions that TIC can well explain generalization gaps.
- We use TIC as a threshold for pruning poorly performing trial models during hyperparameter optimization (HPO) and show that it can successfully prevent promising candidates from being pruned prematurely.

## 2 GENERALIZATION MEASURES

*Generalization measures* measure the generalization ability of statistical models. Typically, the generalization gap, which is defined as the difference between training loss and validation loss, is used to quantify the generalization ability.

### 2.1 WHICH GENERALIZATION MEASURE IS PROSPECTIVE?

To answer this question, before demonstrating the effectiveness of TIC, we highlight the development of research in this area and the motivation behind this work. For understanding generalization behavior, there are two major approaches by quantifying *generalization bounds* and *complexity measures*.

Approach of quantifying *generalization bounds* is pursued by theoretical groups to prove the bound of the generalization gap (Dziugaite & Roy, 2017). Although tight bounds can be proven, they are often based on assumptions that do not apply to practical DNN settings. In addition, no bounds have been shown to describe the performance of the current DNNs to a satisfactory level.

On the other hand, approach of quantifying *complexity measures*, which do not necessarily certify bounds, follows the principle of Occam's razor and evaluates the complexity of the model. Theoretically motivated complexity measures, including VC-dimension (Vapnik & Chervonenkis, 2015), PAC-Bayesian framework (McAllester, 1999), the norm of parameters (Neyshabur et al., 2015), are often discussed as significant components of generalization bounds, and a monotonic relationship between complexity measures and generalizations is mathematically established. In contrast, empirically motivated generalization measures, such as sharpness (Keskar et al., 2016), are justified by experiments and observations. In particular, for DNNs, Jiang et al. (2019) have conducted exhaustive experiments to evaluate the effectiveness of generalization measures for three groups: norm-based measure, sharpness-based measure, and noise-based measure.

- **Norm-based measure: $|\boldsymbol{\theta}|$.** Most of the proposed norm-based measures are based on the Fisher-Rao Metric (Liang et al., 2019), which does not capture generalization well. In particular, it has been reported that spectral complexity such as product of spectral norms of the layers (Bartlett et al., 2017) has a strong negative correlation with generalization. It is impossible to explain the success of DNN models with huge parameter sizes in recent years with these metrics.

- **Sharpness-based measure: $\boldsymbol{H}(\boldsymbol{\theta})$.** Sharpness-based metrics, such as sharp minima and flat minima (Keskar et al., 2016) and PAC-Bayesian framework (McAllester, 1999), are not only associated with intuitive understanding but also empirically show a strong correlation with the generalization gap. However, some model architectures are known to show poor correlation (Dinh et al., 2017).

- **Noise-based measure: $\boldsymbol{C}(\boldsymbol{\theta})$.** Experimental results show that generalization measure based on gradient has potential (Jiang et al., 2019). In particular, in their experiments, they observe that the variance of the gradient captures Sharpness, but they suggest that this is not a good generalization measure depending on the architecture of the model.

These results suggest that studying generalization measures that can be estimated using $\boldsymbol{H}(\boldsymbol{\theta})$ and $\boldsymbol{C}(\boldsymbol{\theta})$ is prospective. However, since the combination of $\boldsymbol{H}(\boldsymbol{\theta})$ and $\boldsymbol{C}(\boldsymbol{\theta})$ seen in TIC is not feasible to compute for practical DNN settings, so it was outside the scope of (Jiang et al., 2019).

## 2.2 INFORMATION MATRIX: ELEMENTS OF GENERALIZATION MEASURES

Previous research has highlighted information matrices such as $\boldsymbol{H}(\boldsymbol{\theta})$ and $\boldsymbol{C}(\boldsymbol{\theta})$ in generalization measures in DNNs. Thomas et al. (2020); Kunstner et al. (2019) remarked that these matrices are often confused and misused, for example in the field of optimization, leading to wrong conclusions, even though these matrices play an essential role in the study of DNNs, especially in optimization (Amari et al., 2020; Martens & Grosse, 2015a), understanding implicit regularization in SGD (Wen et al., 2019; Zhu et al., 2019), and Bayesian inference (Zhang et al., 2018). Before discussing these generalization measures, it should be made clear how each of the information matrices are defined.

In this paper, uncentered gradient covariance matrix is denoted as $\boldsymbol{C}(\boldsymbol{\theta})$. We define $q_{\boldsymbol{\theta}}$ as a ***model distribution***. Furthermore, we employ the ***data distributions*** $\hat{p}$ and $p$ introduced in the previous section as the empirical and true data distributions respectively. Matrices $\boldsymbol{H}(\boldsymbol{\theta})$, $\boldsymbol{C}(\boldsymbol{\theta})$ and $\boldsymbol{F}(\boldsymbol{\theta})$ are then defined as:

$$\boldsymbol{H}(\boldsymbol{\theta}) = \mathbb{E}_p \left[ \frac{\partial^2 l(y, f(x, \boldsymbol{\theta}))}{\partial \boldsymbol{\theta} \partial \boldsymbol{\theta}^T} \right] \in \mathbb{R}^{d \times d},$$

$$\boldsymbol{C}(\boldsymbol{\theta}) = \mathbb{E}_p \left[ \frac{\partial l(y, f(x, \boldsymbol{\theta}))}{\partial \boldsymbol{\theta}} \frac{\partial l(y, f(x, \boldsymbol{\theta}))}{\partial \boldsymbol{\theta}^T} \right] \in \mathbb{R}^{d \times d}, \tag{2}$$

$$\boldsymbol{F}(\boldsymbol{\theta}) = \mathbb{E}_{q_{\boldsymbol{\theta}}} \left[ \frac{\partial l(y, f(x, \boldsymbol{\theta}))}{\partial \boldsymbol{\theta}} \frac{\partial l(y, f(x, \boldsymbol{\theta}))}{\partial \boldsymbol{\theta}^T} \right] \in \mathbb{R}^{d \times d}$$

The conditions under which these matrices are equal will be discussed in detail in section 3.1. The relation between $\boldsymbol{C}(\boldsymbol{\theta})$ and $\boldsymbol{F}(\boldsymbol{\theta})$ is often misunderstood because they both involve the outer product of the gradients but they a different distribution when computing the expectation.

As a subsequent study, Novak et al. (2018a) concluded that consideration of either $\boldsymbol{H}(\boldsymbol{\theta})$ or $\boldsymbol{C}(\boldsymbol{\theta})$ alone is insufficient to estimate the generalization of DNNs and that both are essential. In particular, $\boldsymbol{H}(\boldsymbol{\theta})$ is a value that does not depend on the distribution of input data; however, as the generalization ability depends on the distribution of the data, it is also natural to consider $\boldsymbol{C}(\boldsymbol{\theta})$, which is related to noise in the gradient. Furthermore, as supporting evidence of Novak et al. (2018a)'s claim, Thomas et al. (2020) showed empirically the effectiveness of TIC, a generalization measure that considers both $\boldsymbol{H}(\boldsymbol{\theta})$ and $\boldsymbol{C}(\boldsymbol{\theta})$ expressed by the equation 1. However, Thomas et al. (2020)'s work only experimented with very small-scale NNs because it is challenging to calculate TIC with DNNs of practical size. As a matter of fact, even the ResNet-8 model used in the small-scale image classification benchmark CIFAR10 is not feasible, as it requires nearly 200TB of memory to calculate the TIC exactly.

**Remark 2.1.** It should also be noted that TIC is an information criterion for regular models, not for singular models such as DNNs, and its theoretical justification in the domain of DNNs is not clear. One of the characteristics of singular models is that the Fisher information matrix is not positive definite.

## 2.3 TIC IS DERIVED AS GENERALIZATION GAP IN NTK REGIME

This section outlines the derivation of the definition of TIC in equation 1, considering the generalization gap of DNNs in the framework of NTK's regime. We employ the setting introduced in section 1, $f$ is a smooth function over $\boldsymbol{\theta} \in \mathbb{R}^d$, a parameter of the statistical model. First, we further assume that the following holds for $f$ and $\boldsymbol{\theta} \in \mathbb{R}^d$ in the NTK regime.

**Assumption 2.1.**

*(A1)* Global convergence: the model has only one possible solution. However, it is not required to be $q_{\boldsymbol{\theta}} = p$ (allowing for misspecified situation).

*(A2)* Asymptotic normality: the maximum likelihood estimator $\hat{\boldsymbol{\theta}}$ from the empirical data distribution $\hat{p}$ and the maximum likelihood estimator $\theta^*$ in the true data distribution $p$ satisfy asymptotic normality.

**Proposition 2.1** (**Generalization Gap in NTK Regime is equal to TIC**).

Under the assumptions (A1) and (A2), the estimated bias $b$ (*i.e.* generalization gap) when evaluating using empirical data distribution $\hat{p}$ would then be as follows.

$$
\begin{aligned}
b &= \mathbb{E}_p \left[ \mathbb{E}_{\hat{p}}[l(\boldsymbol{y}, f(\boldsymbol{x}, \hat{\boldsymbol{\theta}}))] - \mathbb{E}_p[l(\boldsymbol{y}, f(\boldsymbol{x}, \hat{\boldsymbol{\theta}}))] \right] \\
&= \mathrm{Tr}\left( \boldsymbol{H}_p(\boldsymbol{\theta}^*)^{-1} \boldsymbol{C}_p(\boldsymbol{\theta}^*) \right)
\end{aligned}
\tag{3}
$$

Where $\boldsymbol{H_p}(\boldsymbol{\theta}^*)$ and $\boldsymbol{C_p}(\boldsymbol{\theta}^*)$ are the Hessian and Covariance respectively with regards to $\theta^*$ over true data distribution $p$. However, as the true data distribution $p$ and parameter $\theta^*$ which maximizes the likelihood for that data distribution are unknown, the expected value in the empirical data distribution $\hat{p}$ and parameter $\hat{\theta}$ are generally used as a consistent estimator, which is consistent with the TIC. A more detailed proof is given in Appendix A.1.2.

**Remark 2.2.** The bias term of TIC is formulated as $\mathrm{Tr}\left( \boldsymbol{H}(\boldsymbol{\theta})^{-1} \boldsymbol{C}(\boldsymbol{\theta}) \right)$. However, since there is no guarantee that $\boldsymbol{H}(\boldsymbol{\theta})$ is positive definite in practice. To prevent this problem, the addition of a small identity matrix, called damping, is performed as $\tilde{\boldsymbol{H}}(\boldsymbol{\theta})^{-1} = (\boldsymbol{H}(\boldsymbol{\theta}) + \lambda I)^{-1}$. Alternatively, consider the case where the TIC is calculated by approximation with a matrix of only the diagonal components of the respective matrices, as $\mathrm{Tr}\left( \boldsymbol{H}(\boldsymbol{\theta})^{-1} \boldsymbol{C}(\boldsymbol{\theta}) \right) \approx \mathrm{Tr}\left( \boldsymbol{H}_{\mathrm{diag}}(\boldsymbol{\theta})^{-1} \boldsymbol{C}_{\mathrm{diag}}(\boldsymbol{\theta}) \right)$ In this case, the following lower bound is given for the diagonal approximated TIC.

$$
\mathrm{Tr}\left( \boldsymbol{H}_{\mathrm{diag}}(\boldsymbol{\theta})^{-1} \boldsymbol{C}_{\mathrm{diag}}(\boldsymbol{\theta}) \right) > \frac{\mathrm{Tr}(\boldsymbol{C}_{\mathrm{diag}}(\boldsymbol{\theta}))}{\mathrm{Tr}(\boldsymbol{H}_{\mathrm{diag}}(\boldsymbol{\theta}))} = \frac{\mathrm{Tr}(\boldsymbol{C}(\boldsymbol{\theta}))}{\mathrm{Tr}(\boldsymbol{H}(\boldsymbol{\theta}))}
\tag{4}
$$

**Remark 2.3.** We note that not all DNNs are in the NTK regime. One indicator of whether a DNN is in the NTK regime is the ratio of the number of model parameters to the number of data. In general, unconstrained DNNs are singular models, so WAIC (Watanabe, 2013) is appropriate instead of TIC or AIC (Akaike, 1998), but computational cost of WAIC is way too high to perform a wide range of learning experiments. Furthermore, when the loss function includes a regularization term, GIC (Konishi & Kitagawa, 1996) is technically appropriate instead of TIC, but it has a disadvantage that the calculation is further complicated.

## 3 APPROXIMATION OF TIC

### 3.1 HESSIAN, GENERALIZED GAUSS-NEWTON MATRIX (GGN) AND FIM

In this section, we describe the conditions for which the Hessian, GGN, and FIM become equivalent. This equivalence can be exploited to reduce the computational cost of computing the TIC. TIC requires the computation of $\boldsymbol{H}(\boldsymbol{\theta})$ and $\boldsymbol{C}(\boldsymbol{\theta})$, but the computational cost of $\boldsymbol{H}(\boldsymbol{\theta})$ is relatively high. For NNs that consist of linear, convolutional, and pooling layers, along with piecewise linear activations, the Hessian is equal to the GGN (Schraudolph, 2002). This actually holds true for most CNNs used in practice. The GGN is an extension of the Gauss-Newton matrix $\tilde{\boldsymbol{G}}(\boldsymbol{\theta}) = \mathbb{E}_p \left[ (\boldsymbol{J_\theta})^T \boldsymbol{J_\theta} \right]$.

$$
\boldsymbol{G}(\boldsymbol{\theta}) = \mathbb{E}_p \left[ (\boldsymbol{J_\theta})^T \boldsymbol{H}_f \boldsymbol{J_\theta} \right]
\tag{5}
$$

Where $\boldsymbol{H}_f$ is the Hessian of $l(y, f(x, \boldsymbol{\theta}))$ and $\boldsymbol{J_\theta}$ is Jacobian of $f(x, \boldsymbol{\theta})$ with respect to $\boldsymbol{\theta}$. Furthermore, the GGN is equal to the FIM for any NN that uses the softnax cross entropy. Therefore, we can assume the following for most practical DNN problem settings.

**Assumption 3.1.**

    *(B1)* Loss function: $l$ is the softmax cross-entropy function

    *(B2)* Activation function: inside $f$, all activation functions' second derivative are always zero, such as ReLU or the identity function.

**Proposition 3.1 ($\boldsymbol{H(\theta)}$ is equal to $\boldsymbol{F(\theta)}$ through GGN).**

Under the assumption of (B1) and (B2), $\boldsymbol{H(\theta)}$ and $\boldsymbol{F(\theta)}$ are exactly equal through GGN. They are also guaranteed to be positive semi-definite.

$$\boldsymbol{H(\theta)} = \boldsymbol{G(\theta)} = \boldsymbol{F(\theta)} \tag{6}$$

A more detailed proof is given in Martens (2020).

### 3.2 Approximation of Matrices and Trace Estimation

As noted in section 2.2, information matrices are in demand for many applications, including TIC. However, for a model with a large number of parameters $d$ such as a DNN, it is necessary to compute a matrix with size of $d^2$. For this reason, approximation methods ranging from approximating the information matrix itself (Le Roux et al., 2007) to approximating the product of the information matrix, and the vector directly is used in optimization (Pearlmutter, 1994) and other applications. We propose the following approximation method to calculate the TIC and experimentally verify the trade-off between accuracy and computation time.

- **Replacing $\boldsymbol{H(\theta)}$ in $\boldsymbol{F(\theta)}$ and fast estimation of $\boldsymbol{F(\theta)}$ in Monte-Carlo sampling.** As shown in equation 6, $\boldsymbol{F(\theta)}$ can be used in place of $\boldsymbol{H(\theta)}$ under the (B1) and (B2) assumption. We use this property to speed up the calculation by simultaneously computing $\boldsymbol{C(\theta)}$ and $\boldsymbol{F(\theta)}$, which have a common term. Furthermore, since the number of classes for the classification task is 10 in MNIST and 100 in CIFAR100, the computational cost of $\boldsymbol{F(\theta)}$ is huge, so we approximate $\boldsymbol{F(\theta)}$ using $\boldsymbol{F}_{\mathrm{mmc}}(\boldsymbol{\theta})$, which is a Monte Carlo approximation. Martens & Grosse (2015b) use $m = 1$ in the practical setting. We follow this setting $\boldsymbol{F}_{\mathrm{1mc}}(\boldsymbol{\theta})$ for the approximation of $\boldsymbol{F(\theta)}$.

- **Block-diagonalization and diagonalization.** In NTK Regime, the correlation between layers is ignored, so block-diagonalization is a reasonable approximation method. The computational complexity can be reduced from $O(d^3)$ to $O(d_l^3)$[2] by the block-diagonal approximation. Diagonalization is a simple approximation; it ignores the correlation between DNN units. It has been reported to be sufficient for some applications (Singh & Alistarh, 2020). It can also be calculated as a sum-of-products operation on vectors rather than matrices, significantly reducing computational complexity and memory consumption. In particular, by using the diagonal approximation, the inverse calculation of $\boldsymbol{H(\theta)}$ can be reduced from $O(d^3)$ to the order of $O(d)$.

- **Lower bound of diagonalization.** As shown in equation 4, by giving the lower bound of the diagonal approximation, it is possible to calculate the trace of each matrix by calculating and dividing the trace of each matrix without calculating the diagonal component of the matrix. In other words, it is possible to calculate without considering whether $\boldsymbol{F(\theta)}$ is positive definite.

- **Hutchinson's method for estimating $\mathrm{Tr}(\boldsymbol{H(\theta)})$ in fast.** Furthermore, rather than approximating the matrix itself, we will introduce a method to accelerate the computation of its eigenvalues and trace. For optimization in deep learning, it is enough to calculate not the Hessian itself but the product of the Hessian and an arbitrary vector (Hessian vector product; Hvp). In order to compute Hvp exactly, Pearlmutter (1994) proposed a fast algorithm to compute Hvp in NNs during backpropagation. This Hvp can be applied to non-optimization applications, such as approximating $\mathrm{Tr}(\boldsymbol{H(\theta)})$ (Avron & Toledo, 2011). Hutchinson's method (Hutchinson, 1989) approximates the expectation value of the quadratic form of the Hessian and the Rademacher random vector (each element takes 1 or $-1$ with probability $1/2$).

---

[2] $l$ is the number of parameters for the layer with the largest number of parameters in the network.

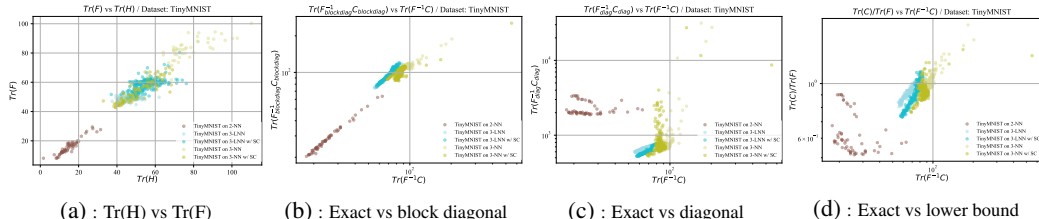

|  |  |  |  |
| :---: | :---: | :---: | :---: |
| (a) : Tr(H) vs Tr(F) | (b) : Exact vs block diagonal | (c) : Exact vs diagonal | (d) : Exact vs lower bound |

Figure 1: **Approximation comparison experiments in small-scale setting.** (a) shows equality of $Tr(H)$ and $Tr(F)$. (b), (c), and (d) respectively compare how different the approximation method of TIC estimation is from the exact case. All full results are shown in Appendix D.2.2.

## 4 EXPERIMENTS

### 4.1 OVERVIEW

The goal of our paper is to elucidate the correlation between the TIC estimates and the generalization gap. To make our study of TIC as comprehensive as possible, we trained on 4 different data sets and 12 different DNN models. Using these combinations, we searched for hyperparameters for each of the 15 problem settings and evaluated the parameters of the trained models. By comparing these results, we can observe how the effectiveness of TIC changes with the model and problem settings. In our experiments, the bias term of TIC is estimated by using validation data, and the generalization gap is the absolute value of the difference in loss between training and test data, using all of the data in each dataset, not just a part of the data. The problem settings for the experiment can be divided into two main categories along with dataset and model size as table 1.

Table 1: **2 Categories of experimental settings.** Problem settings with $\sharp$ and $\star$ indicate to use linear neural network and to be considered almost in NTK regime respectively. For hyperparameter search, we conduct Bayesian optimization for all experimental settings. We describe the further detailed configurations of hyperparameters and other settings for the experiment in Appendix C.2. The remaining experimental settings are explained in Appendix C.

| Category | TIC Estimates | Problem Setting: Dataset & Model | Ratio: $d/n$ |
| :--- | :--- | :--- | :--- |
| **Small-scale** | Exact and Approx. | TinyMNIST on 2-NN w/o SC[3] | 0.09 |
| Data<1MB | (Block Diag, Diag | TinyMNIST on 3-NN w/o and w/ SC | 0.02 |
| Model<50KB | and Lower Bound) | $\sharp$ TinyMNIST on 3-LNN w/o and w/ SC | 0.02 |
| | | $\star$ MNIST on 6-NN w/o and w/ SC | 2.50 |
| | | $\sharp\star$ MNIST on 6-LNN w/o and w/ SC | 2.50 |
| **Practical-scale** | Approx. | $\star$ MNIST on Simple CNN | 268.92 |
| Data>20MB | (Diag and | $\sharp\star$ CIFAR10 on 6-LNN w/o and w/ SC | 8.72 |
| Model>0.5MB | Lower Bound) | $\star$ CIFAR10 on ResNet8 w/o BN [4] | 122.65 |
| | | $\star$ CIFAR10 on VGG16 w/o BN | 3357.53 |
| | | $\star$ CIFAR100 on ResNet8 w/o BN | 122.65 |

In particular, ResNet-8, which is commonly used as a benchmark for training CIFAR10, requires over 200TB of memory to compute exact $\boldsymbol{H}(\boldsymbol{\theta})$. Even state-of-the-art GPU NVIDIA A100 is impractical since it has only 80 GB of device memory. Hence, as **small-scale** experiment, we use a small dataset called TinyMNIST to limit the size of the DNN model, which is a resized version of the MNIST image, which reduces the dimension of the input layer of DNN, to compare our approximation method and exact calculation. As **practical-scale** experiments, we evaluated the real-world datasets and DNN models. We used diagonal approximations and their lower bound approximations to estimate TIC.

### 4.2 SMALL SCALE EXPERIMENTS: COMPARING APPROXIMATION AND EXACT

As **small-scale** experiments, we trained Tiny MNIST on 5 experimental settings: 3-LNNs and NNs, each w/ and w/o SC, and a wide model, 2-NNs without SC. Afterwards, we evaluated the approximation of $\mathrm{Tr}\left(\boldsymbol{H}(\boldsymbol{\theta})^{-1}\boldsymbol{C}(\boldsymbol{\theta})\right)$, the bias term of TIC, for $\boldsymbol{H}(\boldsymbol{\theta})$ and $\boldsymbol{C}(\boldsymbol{\theta})$, using block-diagonal

---

[3]2-NN and 3-LNN denote 2-Layer Non Linear Neural Network and 3-Layer Linear Neural Network respectively. SC denotes Skip-Connection

[4]BN denotes Batch Normalization.

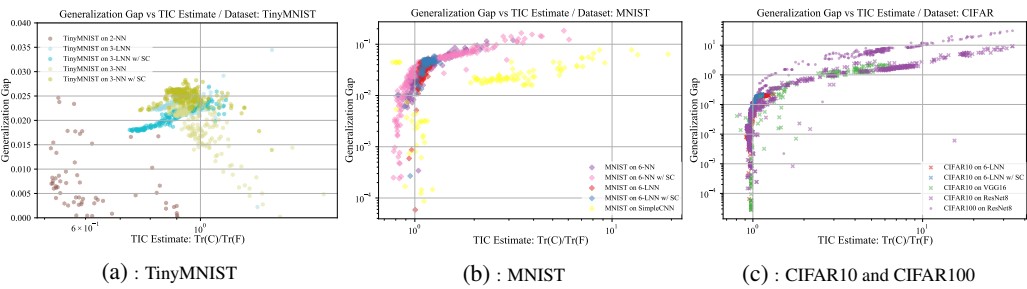

Figure 2: **Correlation between the generalization gap and the TIC estimates.** (a) is a problem setting outside the NTK regime, where the correlation between TIC and the generalization gap is weak; (b) (c) are a problem setting close to the NTK regime, where the correlation is stronger. All full results are shown in Appendix D.

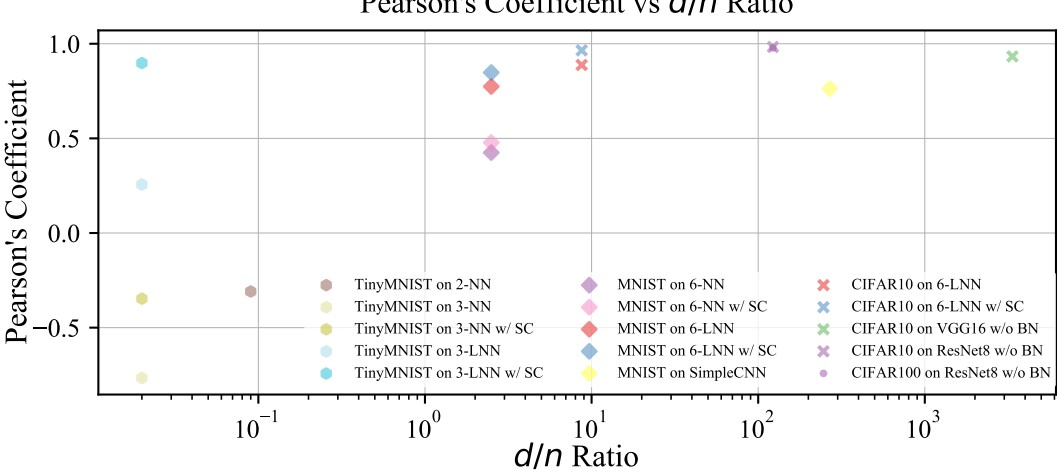

Figure 3: **Relationship between Pearson's Correlation (generalization gap and TIC estimates), and** $d/n$. It should be noted that the correlation between the TIC estimates and the generalization gap is high in regions with large $d/n$, which are considered to be close to the NTK regime. All full results of its value and other metric's Spearman's Correlation and Kendall's $\tau$ Coefficient result including are shown in Appendix D.

approximation, diagonal approximation, and its lower bound. Furthermore, as mentioned in equation 6, for the purpose of speeding up the process, we also estimate TIC using $\boldsymbol{F}(\boldsymbol{\theta})$, which shares the same elements to be calculated as $\boldsymbol{C}(\boldsymbol{\theta})$ as an alternative.

**Remark 4.1.** It should be noted that the above five settings are different from the situation of NTK, since $d \ll p$. However, we observed that the estimation of TIC was effective for LNNs.

First, we show the results of our experiments on the quality of the approximations. In general, from the exact computation to the block-diagonal approximation, i.e., the approximation which ignores the correlation between layers, we can confirm that the value and the rank correlation are kept. As for the LNN, the rank correlation is maintained for the block-diagonal approximation, the diagonal approximation, and its lower bound, though the value fluctuates. On the other hand, in the case of NN w/ SC, we confirmed that the rank correlation is maintained between exact and block-diagonal approximation, between diagonal approximation and its lower bound. These results show that LNN or NN with more layers and SC has a trend of the higher approximation quality.

Then we explain the correlation between the TIC estimates and the generalization gap. We observed that LNN is in the effective regime of TIC and has a high correlation with the generalization gap in all approximations. For NNs, similarly high correlations were observed for the models w/ SC. For the 3-NN w/o SC, the results were such that the inverse correlation was observed even in the exact case. In the case of 2-NN, the approximate correlation was also collapsed, resulting in no correlation with the generalization gap.

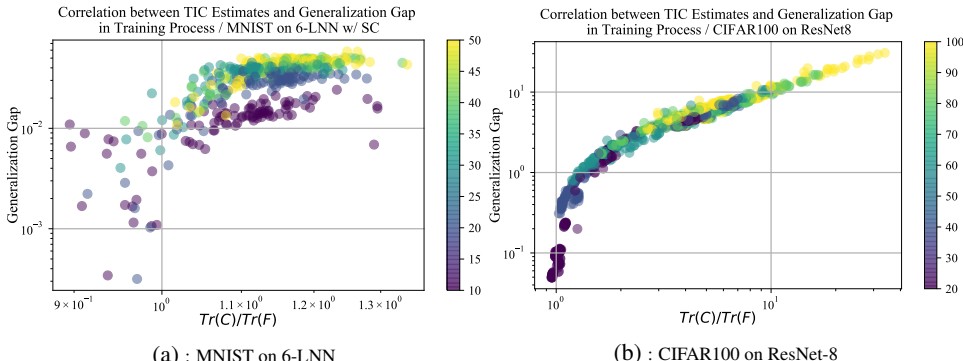

(a) : MNIST on 6-LNN          (b) : CIFAR100 on ResNet-8

Figure 4: **Correlation between the generalization gap and the TIC estimates in training process**. The color bar represents the number of epochs of trained models. All full results are shown in Appendix D.3

From these results, we conclude that the performance of TIC on the correlation with the generalization gap is higher for NN models with more layers and SC, and the correlation does not change significantly before and after the approximation.

### 4.3 PRACTICAL SCALE EXPERIMENTS: CORRELATION TO GENERALIZATION GAP AND TIC LOWER BOUND, TIC WITH DIAGONAL APPROXIMATION

As **practical-scale** experiments, we experimented with the problems where $d \gg p$, which is considered to be NTK's Regime. Contrary to small-scale experiments, we used MNIST, CIFAR10, and CIFAR100 datasets to evaluate practical settings.

First, we show the case of MNIST. The settings of LNN show a strong correlation with the generalization gap in the lower bound approximation as well as in the small-scale experiment. In the case of the NN model, a strong correlation with the generalization gap is observed, unlike in the small-scale setting. Furthermore, in the case of NN and LNN w/ SC, it has less variance and shows a stronger correlation with the generalization gap. In the Simple CNN case, the correlation with the generalization gap is weaker than in previous cases but still shows a correlation. Also, there is no correlation with the generalization gap in the case of the value of $\mathrm{Tr}\,(\boldsymbol{H}(\boldsymbol{\theta}))$, $\mathrm{Tr}\,(\boldsymbol{F}(\boldsymbol{\theta}))$, $\mathrm{Tr}\,(\boldsymbol{C}(\boldsymbol{\theta}))$ itself respectively. Detailed experimental results are shown in figure 13 in the Appendix D.3.

In the cases of CIFAR10 and CIFAR100, both the measures using lower bound and the diagonal approximation show a high correlation with the generalization gap. For LNNs, the correlation is more linear in the case w/ SC. For VGG16 and ResNet8, the correlation is not as good as for LNN, but we confirmed the effectiveness of TIC in NTK's regime. Furthermore, no correlation was found between the generalization gap and trace itself, respectively. These trace values have different patterns depending on the network, and it was found that this single factor alone is insufficient for estimating the generalization gap.

**Remark 4.2.** It should be noted that TIC estimates captured the trend of the generalization gap in the training process as shown in figure 4.

### 4.4 CALCULATION RUNTIME MEASUMENT EXPERIMENTS

Our runtime measurement experiments were run on an NVIDIA Tesla V100 16GB GPUs, with an average of 10 trials each. Significant speedup was achieved by approximating the shape of the matrix, replacing $\boldsymbol{H}(\boldsymbol{\theta})$ by $\boldsymbol{F}(\boldsymbol{\theta})$, and Monte Carlo estimation of $\boldsymbol{F}(\boldsymbol{\theta})$, as shown in 3. Even in the case of a small-scale problem setting, the diagonal approximation with $\boldsymbol{F}(\boldsymbol{\theta})$ and $\boldsymbol{C}(\boldsymbol{\theta})$ is 50 times faster than the exact version, while maintaining the rank correlation with $\boldsymbol{H}(\boldsymbol{\theta})$ and $\boldsymbol{C}(\boldsymbol{\theta})$. However, since the number of parameters in the small problem setting is at most 720, and VGG16 has 186,530 times as many parameters, the effect of increasing the computational order from $O(d^3)$ to $O(d)$ is more significant in the large-scale problem setting. The full details are shown in Appendix D.4. Notably, this speedup by using $\boldsymbol{F}(\boldsymbol{\theta})$ and $\boldsymbol{C}(\boldsymbol{\theta})$ as a set instead of $\boldsymbol{H}(\boldsymbol{\theta})$ and $\boldsymbol{C}(\boldsymbol{\theta})$, and the method of approximating the matrix form to drop the computation order dramatically reduces the computation time.

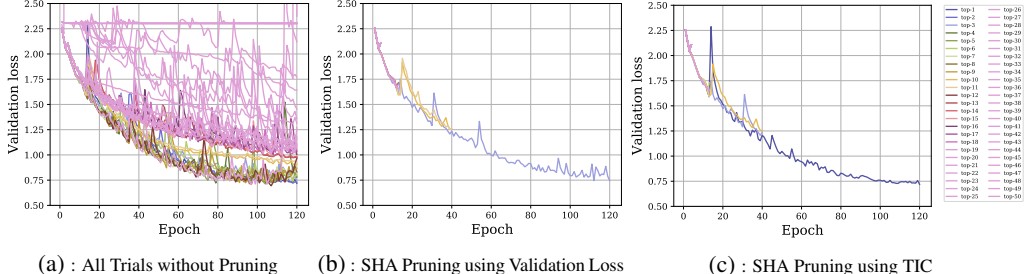

(a) : All Trials without Pruning     (b) : SHA Pruning using Validation Loss     (c) : SHA Pruning using TIC

Figure 5: **A comparative experiment using TIC as an evaluation value for pruning with SHA in HPO for training of CIFAR10 on ResNet-8**: (a) shows the case where all hyperparameter candidates are trained to the end without pruning. (b) shows the case where pruning is performed based on validation loss as a baseline. (c) shows the pruning method using TIC. In the figure, all the legends on the right side show the trials with different hyperparameters, and the final generalization performance (validation loss) to be reached is in descending order. The 1st place trial is shown in dark purple and the 3rd place in light purple.

## 5 Application to Hyperparameter Optimization

In previous sections, we have demonstrated that TIC is a reasonable estimator of the generalization gap that is also effective in the training process and can be computed fast. Motivated by these, in this section, we employ the TIC values on the training processes to accelerate hyperparameter optimization (HPO). HPO is an essential task to achieve good performance in a wide range of machine learning algorithms (Feurer & Hutter, 2019). In particular, the performance of DNNs depends significantly on the selection of the hyperparameters, such as learning rates, weight decay, and momentum (Lucic et al., 2018; Henderson et al., 2018; Dacrema et al., 2019).

The Successive halving algorithm (SHA) (Jamieson & Talwalkar, 2016) shows promising performance in HPO by utilizing the iterative structure of DNNs. SHA prunes unpromising hyperparameters at early stage by utilizing not only a final loss but also losses in training process. The validation loss obtained by the hold out method is usually used as the intermediate loss for SHA. However, the validation loss is often numerically unstable, as shown in Figure 5.

To achieve stable optimization in SHA, we apply the TIC values for the intermediate loss. One advantage of using TIC is that it can take into account the variance as bias term in equation 1, which was not taken into account in the validation loss obtained by the hold out method. In particular, TIC is known to be asymptotically close to leave-one-out cross-validation (LOOCV) (Stone, 1977), and is superior to Hold-Out in terms of the order of estimates error. Details are given in Appendix B. We conduct an experiment to investigate the effectiveness of using the TIC values in SHA. Figure 5 shows the result of the experiment. The TIC values with the proposed approximation method can select 1st top trial, while the traditional method (SHA + the validation loss obtained by the hold out method) selects 3rd top trial.

## 6 Conclusion and Discussion

This study conducted a comprehensive experiment and observed that the TIC approximation method captures the generalization gap, even in the practical DNN setting close to the NTK regime. We have shown that the generalization gap could be captured in the training process, even if the model is not completed to train. Based on these results, we tested the validity of using TIC as an assessment value for HPO branch pruning and confirmed a valid case. It is challenging to establish a theory to discuss in the active regime (outside of the NTK regime) for future work. Especially, WAIC can handle singular models and applied to DNN; the difficulty arises for calculation; thus, it is required to have an approximation method. Still, it is necessary to bridge the theory of DNN for the validity of the approximation.

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

APPENDIX

## A PROOFS

### A.1 DERIVATION OF THE TIC IN NTK REGIME

#### A.1.1 NTK: NEURAL TANGENT KERNEL

In general, DNNs have a large number of parameters $p$, compared to the number of data, $n$, causing them to memorize data and hurt their generalization ability. Furthermore, all data are subjected to nonlinear transformations, which results in the problem of minimizing a nonconvex objective function. Due to this difficulty in analyzing the training dynamics of DNNs, the reasons for the generalization of practical DNNs and the mysteries related to the guarantee of global convergence remain unsolved. However, a theoretical framework called Neural Tangent Kernel (NTK) (Jacot et al., 2018) has been developed to analyze the training dynamics of gradient descent in DNNs with a sufficiently large width.

Assuming that the loss function to be minimized in NN training is $L = \frac{1}{2}\|f(x; \boldsymbol{\theta}) - y\|^2$, and the parameters are updated by the gradient method with the learning rate $\eta$, the amount of change is $\Delta\boldsymbol{\theta}$,

$$\Delta\boldsymbol{\theta} = -\eta\frac{\partial L(\boldsymbol{\theta})}{\partial\boldsymbol{\theta}} = -\eta\nabla_{\boldsymbol{\theta}}L(\boldsymbol{\theta}) \tag{7}$$

Continuing this equation using the continuous time (training step of NN) $t$, we obtain

$$\frac{\partial\boldsymbol{\theta}}{\partial t} = -\eta\nabla_{\boldsymbol{\theta}}L(\boldsymbol{\theta}(t)) \tag{8}$$

$$= -\eta J_t(x; \boldsymbol{\theta})^T\nabla_{f_t(x)}L(f_t(x)) \tag{9}$$

where $J_t(x; \boldsymbol{\theta})$ is $\frac{\partial f(x;\boldsymbol{\theta})}{\partial\boldsymbol{\theta}} = \nabla_{\boldsymbol{\theta}}f_t(x)$.

Second, considering the time variation of the output of the function instead of the parameters we get

$$\frac{\partial f_t(x; \boldsymbol{\theta})}{\partial t} = \frac{\partial f_t(x; \boldsymbol{\theta})}{\partial\boldsymbol{\theta}}\frac{\partial\boldsymbol{\theta}}{\partial t} \tag{10}$$

$$= -\eta J_t(x; \boldsymbol{\theta})J_t(x; \boldsymbol{\theta})^T\nabla_{f_t(x)}L(f_t(x)) \tag{11}$$

$$= -\eta\mathcal{K}_t(x, x)\nabla_{f_t(x)}L(f_t(x)) \tag{12}$$

where $\mathcal{K}_t(x, x)$ is $J_t(x; \boldsymbol{\theta})J_t(x; \boldsymbol{\theta})^T = \sum_{\text{layer}=1}^{\text{num of layer}} J_t(x; \boldsymbol{\theta}_{\text{layer}})J_t(x; \boldsymbol{\theta}_{\text{layer}})^T$

The problem here is that $\mathcal{K}_t(x, x)$ which is the NTK at time $t$ depends on $\boldsymbol{\theta}$ and $x$. However, it was shown that if the width of the randomly initialized NN is large enough, it will be $\mathcal{K}_t(x, x) \sim \mathcal{K}_0(x, x)$ (Jacot et al., 2018).

Given the output of the neural network with its first-order Taylor expansion, we have

$$f_t^{\text{lin}}(x; \boldsymbol{\theta}_t) \approx f_0(x; \boldsymbol{\theta}_0) + \nabla_{\boldsymbol{\theta}}f_0(x; \boldsymbol{\theta}_0)^T(\boldsymbol{\theta}_t - \boldsymbol{\theta}_0) \tag{13}$$

The dynamics of training can be expressed in the following way.

$$\boldsymbol{\theta}_t = \boldsymbol{\theta}_0 - \nabla_{\theta}f_0(x)^T\mathcal{K}_0^{-1}\left(I - e^{-\eta\mathcal{K}_0 t}\right)(f_0(x) - y) \tag{14}$$

$$f_t^{\text{lin}}(x) = \left(I - e^{-\eta\mathcal{K}_0 t}\right)y + e^{-\eta\mathcal{K}_0 t}f_0(x) \tag{15}$$

Therefore, NTK theory determines the dynamics of gradients in function space by introducing the NTK regime, which allows us to assume that the weights follow a Gaussian process even in training progresses, based on the theory that the random initialized NN can be considered as a Gaussian process when the hidden layers become infinite in the study of Lee et al. (2018); Novak et al. (2018b).

NTK allows us to prove the global convergence of gradient descent, and furthermore, the equivalence between the trained model and the Gaussian process can be used to explain the generalization performance of DNN. Besides, NTK was extended to CNNs (Arora et al., 2019) and RNNs (Yang, 2019) as well as MLPs, and exhaustive experiments (Lee et al., 2019) were conducted.

A.1.2 PRELIMINARIES FOR TIC IN NTK REGIME

TIC requires that the statistical model be a regular model. However, DNNs are generally singular models. The requirements for a regular model are as follows,

- The posterior distribution of the parameters can be approximated by a Gaussian distribution, and the number of samples is sufficiently large (as n increases, the prior distribution is ignored).
- There is only one optimal solution $\hat{\boldsymbol{\theta}}$ for $\arg \max \ell(\boldsymbol{\theta})$.
- $\boldsymbol{H}_p(\boldsymbol{\theta}^*)$ is positive definite.

In machine learning, we often pursue to minimize the negative log-likelihood using it as a loss function. Let $f$ be the predictive distribution of the model over the $p$-dimensional parameters $\boldsymbol{\theta} \in \Theta \subset \mathbb{R}^p$ and the true distribution $g$. We can compare the models by measuring the KL divergence of $f$ and $g$ to see how well $f$ approximates the true distribution $g$ as a predictive distribution.

$$D_{KL}(g, f) = \mathbb{E}_p \left[ \log \frac{g(y|x)}{f(y|x, \boldsymbol{\theta})} \right] \tag{16}$$

$$= \mathbb{E}_p[\log g(y|x)] - \mathbb{E}_p[\log f(y|x, \boldsymbol{\theta})] \tag{17}$$

Since the first term of equation 17 is independent of $\boldsymbol{\theta}$, the model is better if it maximizes the second term, i.e., the higher the mean log-likelihood $\mathcal{L}(\boldsymbol{\theta}) = \mathbb{E}_p[\log f(y|x, \boldsymbol{\theta})]$.

The mean log-likelihood $\mathcal{L}(\boldsymbol{\theta})$ is also an unknown quantity that cannot be calculated as it depends on the true distribution of the data $p$. However, suppose it is possible to obtain a valid estimator of the mean log-likelihood using the empirical distribution of the data $\hat{p}$. In that case, it can be used as a criterion for evaluating the model.

In model selection, we assume an output of DNN model $f(y|x, \boldsymbol{\theta})$, which is estimated by the maximum likelihood method. We can calculate $f(y|x, \hat{\boldsymbol{\theta}})$ by replacing the unknown parameters $\boldsymbol{\theta}$ in the probability distribution with the maximum likelihood estimator $\hat{\boldsymbol{\theta}}$.

$$\hat{\boldsymbol{\theta}} := \arg \max_{\boldsymbol{\theta} \in \Theta} \ell(\boldsymbol{\theta}) \tag{18}$$

Where $\ell(\boldsymbol{\theta}) = \mathbb{E}_{\hat{p}}[\log f(y|x, \boldsymbol{\theta})] = \frac{1}{n} \sum_{i=1}^n \log f(y_i|x_i, \boldsymbol{\theta})$ is the likelihood function over $\boldsymbol{\theta} \in \Theta$.

Let $\boldsymbol{S_n} = \{(x_1, y_1), (x_2, y_2), \dots (x_n, y_n)\}$ be the data observed according to the true data distribution $p$. Let $\hat{p}$ be the empirical distribution based on this $\boldsymbol{S_n}$. By the law of large numbers, the mean of $\mathbb{E}_{\hat{p}}[\log f(y|x, \hat{\boldsymbol{\theta}})] = \frac{1}{n} \sum_{i=1}^n \log f(y_i|x_i, \hat{\boldsymbol{\theta}})$ converges in probability to its expected value $\mathbb{E}_p[\log f(y|x, \hat{\boldsymbol{\theta}})]$ when the number of data $n$ becomes infinitely large.

$$\frac{1}{n} \sum_{i=1}^n \log f(y_i|x_i, \hat{\boldsymbol{\theta}}) \xrightarrow[n \to +\infty]{} \mathbb{E}_p[\log f(y|x, \hat{\boldsymbol{\theta}})] \tag{19}$$

Therefore, the estimator based on the empirical distribution function in equation 19 is a natural estimator of the mean log-likelihood. While $\mathbb{E}_{\hat{p}}[\log f(y|x, \hat{\boldsymbol{\theta}})]$ is a natural estimator of $\mathbb{E}_p[\log f(y|x, \hat{\boldsymbol{\theta}})]$, $\hat{\boldsymbol{\theta}}$ is a model parameter estimated using empirical data $(x_i, y_i) \sim \hat{p}$. The data is also used to evaluate the mean log-likelihood of $f(y|x, \hat{\boldsymbol{\theta}})$ of the model in terms of prediction, although a fair model selection is not possible.

Therefore, it is necessary to evaluate and correct for this bias for fair model selection. The bias of estimating the mean log-likelihood $\mathbb{E}_p[\log f(y|x, \hat{\boldsymbol{\theta}})]$ with equation $\mathbb{E}_{\hat{p}}[\log f(y|x, \hat{\boldsymbol{\theta}})]$ is formulated as follows:

$$b = n\mathbb{E}_p \left[ \mathbb{E}_{\hat{p}}[\log f(\boldsymbol{y}|\boldsymbol{x}, \hat{\boldsymbol{\theta}})] - \mathbb{E}_p[\log f(\boldsymbol{y}|\boldsymbol{x}, \hat{\boldsymbol{\theta}})] \right] \tag{20}$$

$$= n\mathbb{E}_p \left[ \ell(\hat{\boldsymbol{\theta}}) - \mathcal{L}(\hat{\boldsymbol{\theta}}) \right] \tag{21}$$

$$= n\mathbb{E}_p \left[ \ell(\hat{\boldsymbol{\theta}}) - \ell(\boldsymbol{\theta}^*) \right] \tag{22}$$

$$+ n\mathbb{E}_p \left[ \ell(\boldsymbol{\theta}^*) - \mathcal{L}(\boldsymbol{\theta}^*) \right] \tag{23}$$

$$+ n\mathbb{E}_p \left[ \mathcal{L}(\boldsymbol{\theta}^*) - \mathcal{L}(\hat{\boldsymbol{\theta}}) \right] \tag{24}$$

Where $\boldsymbol{\theta}^*$ is the maximum likelihood estimator of $\mathcal{L}(\boldsymbol{\theta})$, thus, equation 21 can be decomposed as equation 22, 23, 24. Also, equation 23 converges to 0 ,since taking the expectation of the left-hand side by $p$ coincides with the right-hand side, and equations 22 and 24 converge to $\frac{1}{2}\text{Tr}\left(\boldsymbol{H}_p(\boldsymbol{\theta}^*)^{-1}\boldsymbol{D}_p(\boldsymbol{\theta}^*)\right)$ in $n \to \infty$, respectively. Technically, this asymptotic validity holds under the regularity condition (White, 1982).

### A.1.3 APPLY NTK REGIME FOR TIC DERIVATION

Now we satisfy the above condition by using the training dynamics of DNN in the NTK regime. Specifically, the regime in NTK satisfies the first condition because it uses a locally linear approximation, as in equation 13, and considers the training of the NN as a Gaussian process. Also, as shown in equation 14, the optimal solution at the $t$-th step is uniquely determined, and the optimization is convex.

The positive definiteness of $\mathcal{K}_t(x, x)$ is proved in Appendix A4 of Jacot et al. (2018), under the assumption of non-polynomial Lipschitz nonlinearity. From the definition of $\mathcal{K}_t(x, x)$, the eigenvalues of Fisher Information Matrix (FIM) are positive definite in NTK Regime because $\mathcal{K}_t(x, x)$ and FIM have a duality that shares eigenvalues.

This condition is true when the DNN is considered to be in the NTK Regime, i.e., when Assumption 2.1 is satisfied.

Figure 6 shows a schematic diagram of the bias term $b$. The matrices $\boldsymbol{H}_p(\boldsymbol{\theta}^*)$ and $\boldsymbol{D}_p(\boldsymbol{\theta}^*)$ is as follows:

$$\begin{aligned} \boldsymbol{D}_p(\boldsymbol{\theta}^*) &= \mathbb{E}_p \left[ \frac{\partial \log f(y|x,\boldsymbol{\theta})}{\partial \boldsymbol{\theta}} \frac{\partial \log f(y|x,\boldsymbol{\theta})}{\partial \boldsymbol{\theta}^T} \Big|_{\boldsymbol{\theta}=\boldsymbol{\theta}^*} \right] \\ \boldsymbol{H}_p(\boldsymbol{\theta}^*) &= \mathbb{E}_p \left[ \frac{\partial^2 \log f(y|x,\boldsymbol{\theta})}{\partial \boldsymbol{\theta}\partial \boldsymbol{\theta}^T} \Big|_{\boldsymbol{\theta}=\boldsymbol{\theta}^*} \right] \end{aligned} \tag{25}$$

If the true distribution $g$ is included in the assumed statistical model $f(\boldsymbol{y}|\boldsymbol{x}, \boldsymbol{\theta})$, then $\boldsymbol{D}_p(\boldsymbol{\theta}^*) = \boldsymbol{H}_p(\boldsymbol{\theta}^*)$ is valid and $b = \text{Tr}(\boldsymbol{I}) = d$, and thus AIC can be derived. In the DNN setting, this assumption does not hold, i.e., we need to use the respective matrices for the misspecified situation.

Since the bias $b$ depends on the true data distribution $p$, it needs to be estimated based on the observed data. Assuming that the consistent estimators for $\boldsymbol{D}_p(\boldsymbol{\theta}^*)$ and $\boldsymbol{H}_p(\boldsymbol{\theta}^*)$ are $C(\hat{\boldsymbol{\theta}})$ and $H(\hat{\boldsymbol{\theta}})$, respectively, the estimates in equation 20 are as follows:

$$\begin{aligned} \boldsymbol{C}(\hat{\boldsymbol{\theta}}) &= \frac{1}{n}\sum_{i=1}^n \frac{\partial \log f(y_i|x_i,\boldsymbol{\theta})}{\partial \boldsymbol{\theta}} \frac{\partial \log f(y_i|x_i,\boldsymbol{\theta})}{\partial \boldsymbol{\theta}^T} \Big|_{\boldsymbol{\theta}=\hat{\boldsymbol{\theta}}} \\ \boldsymbol{H}(\hat{\boldsymbol{\theta}}) &= \frac{1}{n}\sum_{i=1}^n \frac{\partial^2 \log f(y_i|x_i,\boldsymbol{\theta})}{\partial \boldsymbol{\theta}\partial \boldsymbol{\theta}^T} \Big|_{\boldsymbol{\theta}=\hat{\boldsymbol{\theta}}} \end{aligned} \tag{26}$$

Using equation 2, $\hat{b}$ as an estimate of bias $b$ can be described as follows.

$$\hat{b} = \text{Tr}\left(\boldsymbol{H}(\hat{\boldsymbol{\theta}})^{-1}\boldsymbol{C}(\hat{\boldsymbol{\theta}})\right) \tag{27}$$

The term of $b$ is called Moody's effective number of parameters (Moody, 1992). The TIC is derived in equation 1 by estimating the asymptotic bias of the mean log-likelihood with the log-likelihood of the statistical model.

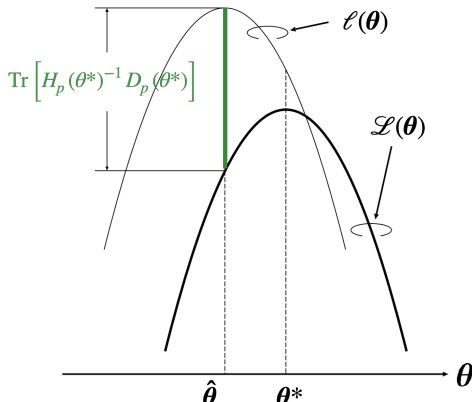

Figure 6: TIC takes into account the bias of the estimate

## B  ASYMPTOTIC EQUIVALENCE OF TIC TO CROSS-VALIDATION

Since deep learning usually requires a lot of data and a huge amount of time for training, the Hold-Out method is commonly used to divide the training data into train data for training, validation data for model selection, especially for hyperparameter optimization, and test data to verify the performance of the model. This method is relatively fast, but its evaluation varies depending on how the data is divided, and it is not used when the number of data is small. In K-fold cross-validation, the entire train data is divided into K pieces. Then, one of them is used as the validation data, and the remaining K-1 groups are decomposed into training data. The validation data and training data are swapped and repeated, and the verification is repeated so that all cases become validation cases. The leave-one-out cross-validation (LOOCV) is a method that uses only one piece of the entire train data as validation data. LOOCV is empirically known to have high performance and is often used when the overall data is small. If the number of data is $n$, the bias of the estimation error is $O(1/\sqrt{n})$ for the Hold-Out method and $O(1/n)$ for LOOCV (Stone, 1977). However, LOOCV requires $n$ times the computational cost. In a case such as ImageNet-1K, 1.2 million images can be used as training data, and the current trend is to use the Hold-Out method, which allows the estimation error to be $O(1/\sqrt{n})$ with small computational cost, instead of reducing the estimation error to $O(1/n)$ at the expense of huge computational cost.

## C  TIC EXPERIMENTAL DETAILS

### C.1  IMPLEMENTATION AND ENVIRONMENT FOR EXPERIMENT

We perform our experiment with supercomputer A (These names will be deanonymized after publication). For supercomputer A, each node is composed of NVIDIA Tesla V100×4GPU and Intel Xeon Gold 6148 2.4 GHz, 20 Cores×2CPU. As a software environment, we use Red Hat 4.8.5, gcc 7.4, Python 3.6.5, Pytorch 1.6.0, cuDNN 7.6.2, and CUDA 10.0. Our code can be found at the link below.
https://anonymous.4open.science/r/TIC-in-NTKRegime-02A2/

### C.2  HYPERPARAMETERS AND DETAILED CONFIGURATION

We will report the hyperparameter's search space. We search learning rate $\eta$, learning rate decay rate $\rho$ and the timing to decay learning rate $\delta$, and regularization coefficient of weight decay $\lambda$. When $\delta = 0.7$, it means that the learning rate decays when training passes 70 % of the total iterations. Furthermore, a parameter to control momentum $\gamma$ is added to the hyperparameters.

To set the range in which to search for each hyperparameter, we follow the configuration of Choi et al. (2019),Shallue et al. (2019). We summarize the workloads we use in the experiment in Table 2. We do not use batch normalization and the input image is just normalized and any data augmentation

is not employed. Hyperparameter ranges are summarized in Tables 3,4,5 and 6. We conduct a bayesian optimization to explore hyperparameters in the range described in these tables.

Table 2: Experiments: Workloads

| Model | Dataset | Batch size | Step Budget | Epoch |
|---|---|---|---|---|
| 2-NN w/o SC (Thomas et al., 2020) | TinyMNIST | 512 | 11343 | 120 |
| 3-LNN w/ SC | TinyMNIST | 8192 | 300 | 60 |
| 3-LNN w/o SC | TinyMNIST | 8192 | 300 | 60 |
| 3-NN w/ SC | TinyMNIST | 8192 | 300 | 60 |
| 3-NN w/o SC | TinyMNIST | 8192 | 300 | 60 |
| 6-LNN w/ SC | MNIST | 8192 | 300 | 60 |
| 6-LNN w/o SC | MNIST | 8192 | 300 | 60 |
| 6-NN w/ SC | MNIST | 8192 | 300 | 60 |
| 6-NN w/o SC | MNIST | 8192 | 300 | 60 |
| Simple CNN Base (Shallue et al., 2019) | MNIST | 256 | 9350 | 60 |
| 6-LNN w/ SC | CIFAR-10 | 256 | 10205 | 60 |
| 6-LNN w/o SC | CIFAR-10 | 256 | 10205 | 60 |
| VGG-16 w/o BN (Simonyan & Zisserman, 2015) | CIFAR-10 | 128 | 78000 | 250 |
| ResNet-8 w/o BN (Shallue et al., 2019) | CIFAR-10 | 256 | 15800 | 120 |
| ResNet-8 w/o BN (Shallue et al., 2019) | CIFAR-100 | 256 | 15800 | 120 |

Table 3: Hyperparameter Search Range for TinyMNIST Dataset Experiments

| Model | $\eta$ | $\rho$ | $\delta$ | $\lambda$ | $\gamma$ |
|---|---|---|---|---|---|
| 2-LNN w/o SC | [1e-3, 1e-1] | [1e-2, 1] | [1e-2, 1] | [0] | [0, 0.999] |
| 3-NN w/o and w/ SC | [1e-4, 1e-1] | [5e-1, 1] | [5e-1, 1] | [0] | [0, 0.999] |
| 3-LNN w/o and w/ SC | [1e-4, 1e-1] | [5e-1, 1] | [5e-1, 1] | [0] | [0, 0.999] |

Table 4: Hyperparameter Search Range for MNIST Dataset Experiments

| Model | $\eta$ | $\rho$ | $\delta$ | $\lambda$ | $\gamma$ |
|---|---|---|---|---|---|
| 6-NN w/o and w/ SC | [1e-4, 1e-1] | [5e-1, 1] | [5e-1, 1] | [0] | [0, 0.999] |
| 6-LNN w/o and w/ SC | [1e-4, 1e-1] | [5e-1, 1] | [5e-1, 1] | [0] | [0, 0.999] |
| Simple CNN | [1e-4, 1] | [5e-1, 1] | [5e-1, 1] | [0] | [0, 0.999] |

Table 5: Hyperparameter Search Range for CIFAR10 Dataset Experiments

| Model | $\eta$ | $\rho$ | $\delta$ | $\lambda$ | $\gamma$ |
|---|---|---|---|---|---|
| 6-LNN w/o and w/ SC | [1e-4, 1e-1] | [0.5, 1] | [0.5, 1] | [0] | [0, 0.999] |
| ResNet-8 w/o BN | [1e-6, 1e+1] | [0.5, 1] | [0.5, 1] | [1e-5, 1e-4] | [1e-4, 0.999] |
| VGG-16 w/o BN | [1e-3, 1e-0] | [0.5, 1] | [0.5, 1] | [1e-4, 1e-1] | [1e-4, 0.999] |

Table 6: Hyperparameter Search Range for CIFAR100 Dataset Experiment

| Model | $\eta$ | $\rho$ | $\delta$ | $\lambda$ | $\gamma$ |
|---|---|---|---|---|---|
| ResNet-8 w/o BN | [1e-6, 1e+1] | [0.5, 1] | [0.5, 1] | [1e-5, 1e-4] | [1e-4, 0.999] |

## C.3 DISTRIBUTION OF TRAIN LOSS AND GENERALIZATION GAP

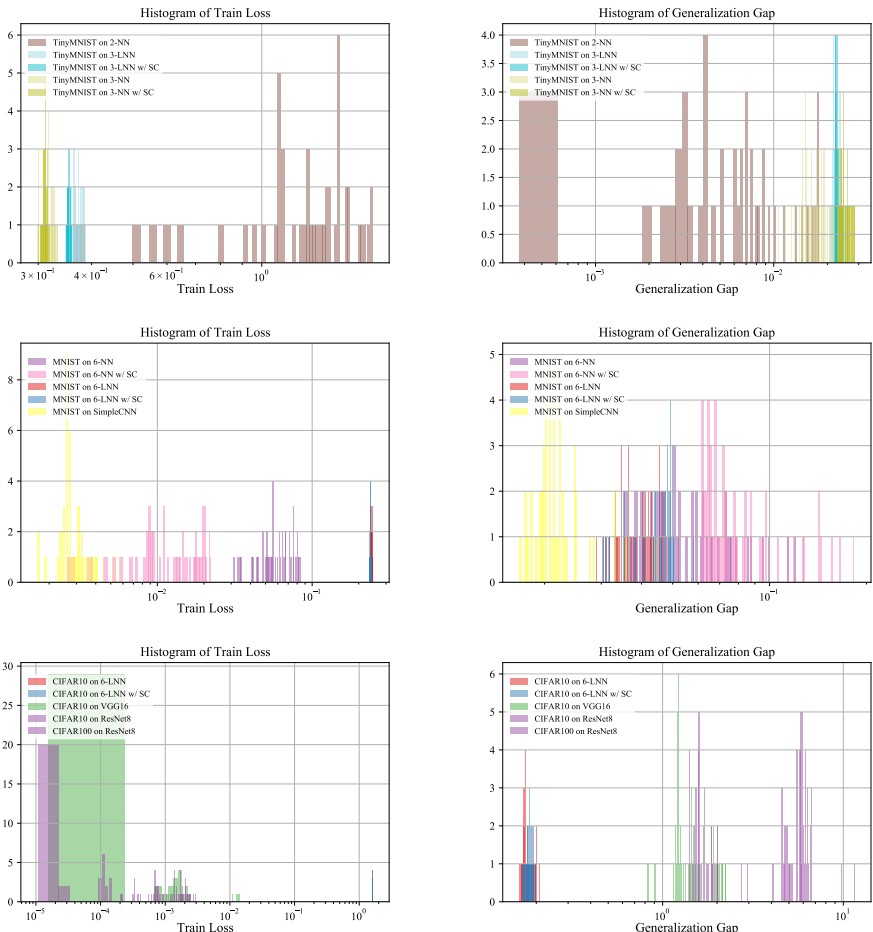

Figure 7: Distribution of training loss and generalization gap on the trained models

# D    ADDITIONAL EXPERIMENTAL RESULTS

## D.1    FULL RESULTS OF CORRELATION BETWEEN GENERALIZATION GAP AND TIC LOWER BOUND ESTIMATES

We summarize these results, evaluated with three different correlation coefficients, in Table 7. These results are the calculated 3 types of correlation coefficients for the plot shown in figure 4a, 4b and 2c . Furthermore, the relationship between these correlation coefficients and the values of the ratios of the parameters to the number of data is shown in Figure 8. The result of plotting these results along with $d/n$ is shown in Figure 8.

Table 7: Correlation: TIC estimates $\mathrm{Tr}(\boldsymbol{C}(\boldsymbol{\theta}))/\mathrm{Tr}(\boldsymbol{F}(\boldsymbol{\theta}))$ and generalization gap

| Model | Dataset | Spearman's | Kendall's $\tau$ | Pearson's Correlation |
|---|---|---|---|---|
| 2-NN | Tiny MNIST | -0.456 | -0.313 | -0.309 |
| 3-NN | Tiny MNIST | -0.631 | -0.44 | -0.766 |
| 3-NN w/ SC | Tiny MNIST | -0.19 | -0.137 | -0.347 |
| 3-LNN | Tiny MNIST | 0.277 | 0.238 | 0.256 |
| 3-LNN w/ SC | Tiny MNIST | 0.932 | 0.795 | 0.898 |
| 6-NN | MNIST | 0.882 | 0.708 | 0.425 |
| 6-NN w/ SC | MNIST | 0.969 | 0.87 | 0.478 |
| 6-LNN | MNIST | 0.682 | 0.465 | 0.774 |
| 6-LNN w/ SC | MNIST | 0.593 | 0.512 | 0.848 |
| Simple CNN | MNIST | 0.923 | 0.553 | 0.763 |
| 6-LNN | CIFAR10 | 0.951 | 0.82 | 0.888 |
| 6-LNN w/ SC | CIFAR10 | 0.976 | 0.877 | 0.965 |
| VGG-16 w/o BN | CIFAR10 | 0.904 | 0.725 | 0.933 |
| ResNet-8 w/o BN | CIFAR10 | 0.912 | 0.766 | 0.983 |
| ResNet-8 w/o BN | CIFAR100 | 0.966 | 0.855 | 0.978 |

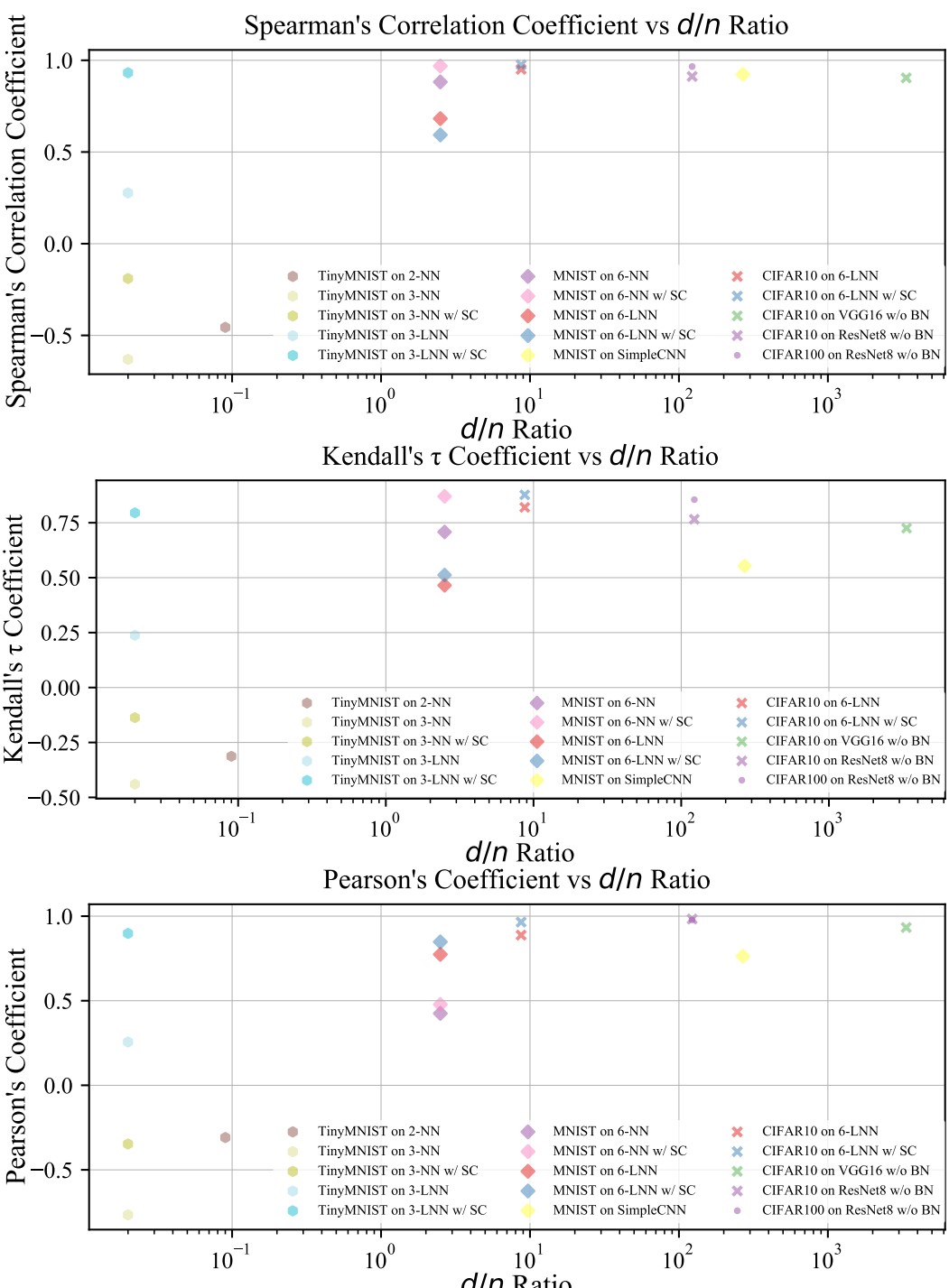

Figure 8: Relationship between correlation coefficient and $d/n$. It should be noted that the correlation between the TIC estimates and the generalization gap is high in regions with large $d/n$, which are considered to be close to the NTK regime.

## D.2    ADDITIONAL RESULTS OF SMALL-SCALE EXPERIMENTS

In this chapter, we provide details of the experimental results on a small-scale that could not be included in the main paper. In particular, we investigate the goodness of approximation of the information matrix in the small-scale case, since it can be computed exactly, although the execution time is longer.

### D.2.1    EMPIRICAL RELATIONSHIP BETWEEN H AND F

First of all, we examine the behavior of $H(\theta)$ when it is approximated by $F(\theta)$, or more precisely, $F_{1mc}(\theta)$. Figure 9 shows that $H(\theta)$ is not in perfect agreement with $F(\theta)$ due to the effect of the damping term. However, For cases that do not require inverse calculations, such as trace calculations and lower bounds, these effects can be eliminated and a relatively good approximation can be achieved. Furthermore, in the case of linear neural networks, $H(\theta)$ and $F(\theta)$ show a strong correlation.

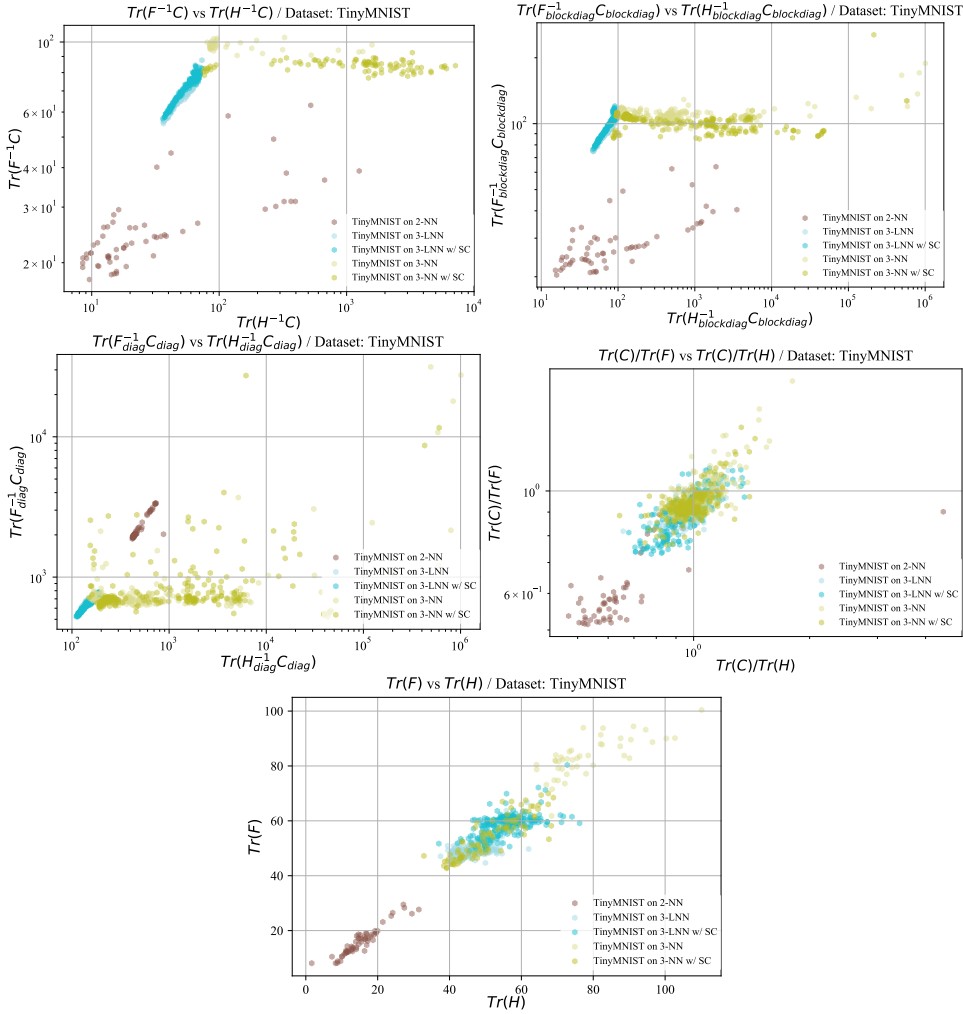

Figure 9: Small-scale experiments: comparison of the TIC estimate w/ $H(\theta)$ and $F(\theta)$ respectively.

### D.2.2 Effect of Matrix Shape Approximation on Estimation of TIC

Next, we fixed the use of $F(\theta)$ instead of $H(\theta)$ and observed the change in the TIC estimates and the correlation with the generalization gap when form of the matrix is changed by approximation methods.

Figure 10 and 11 show the correlation between the TIC estimates and the generalized gaps in the case of approximation. Figure 10 shows the comparison of the shape of matrix and its estimates, and Figure 11 shows the correlation between generalization gap and tic estimates. These values are evaluated using three different correlation metric and summarized in Table 8, 9 and 10.

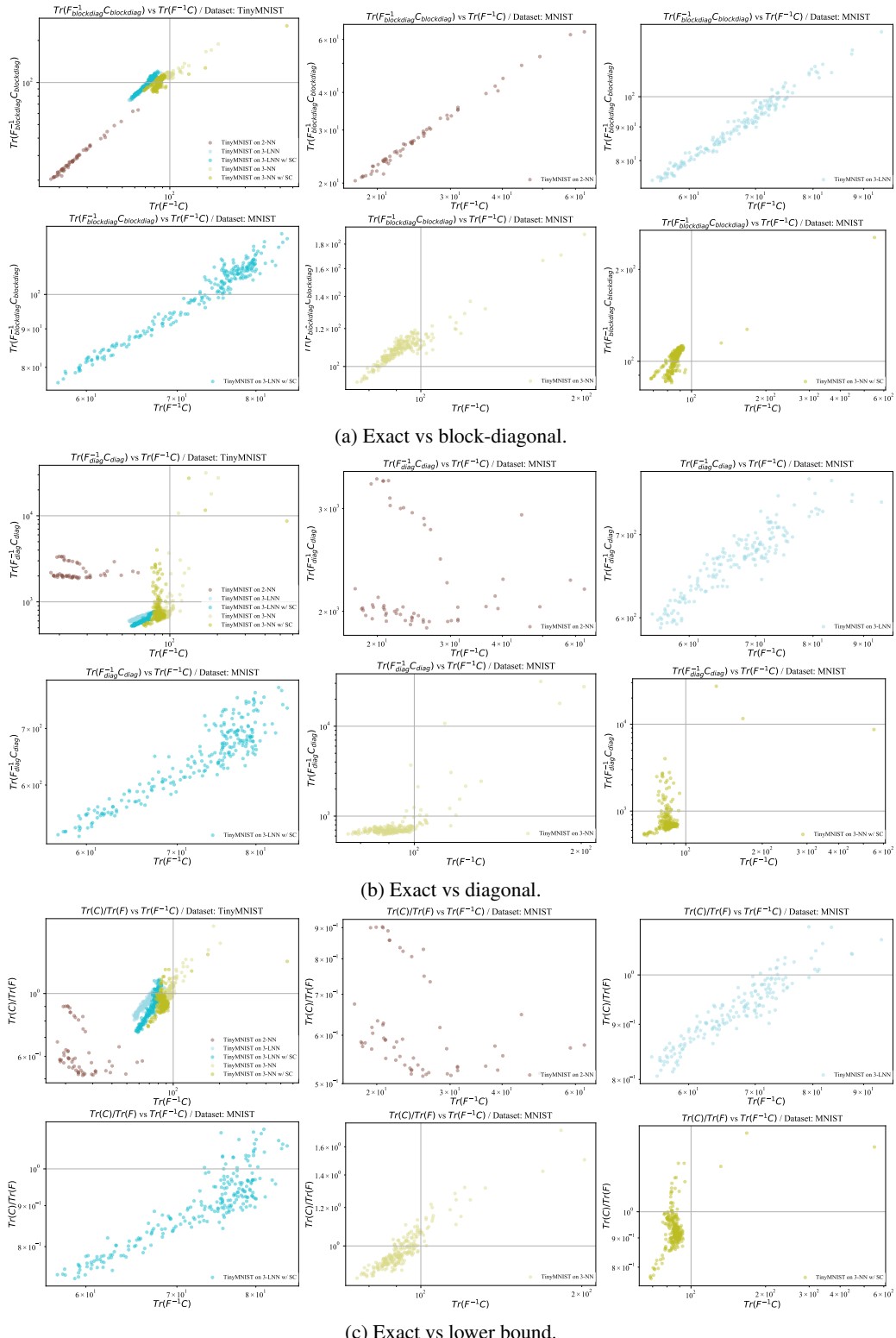

Figure 10: Small-scale experiments: comparison of the TIC estimate.

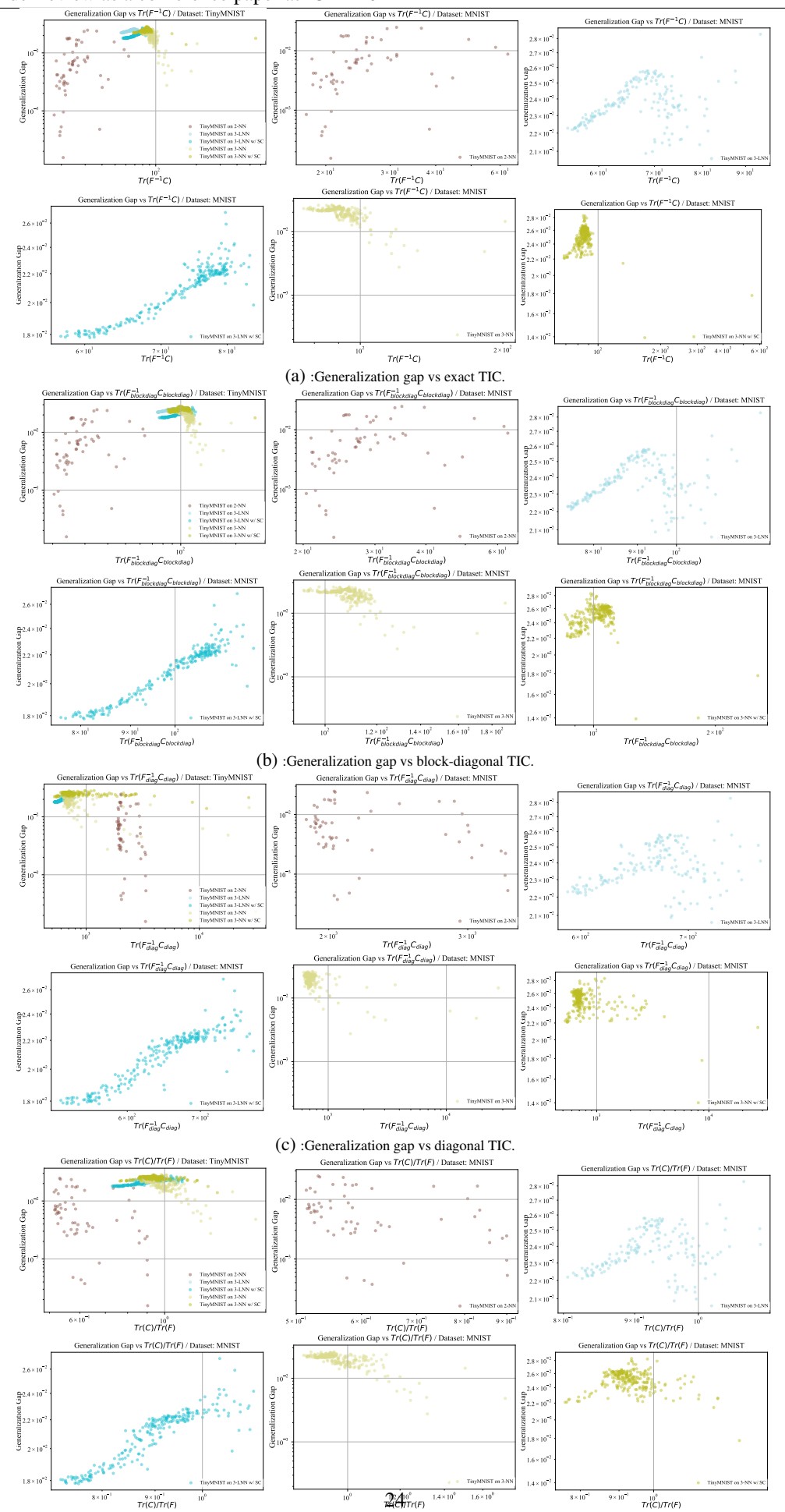

(a) :Generalization gap vs exact TIC.

(b) :Generalization gap vs block-diagonal TIC.

(c) :Generalization gap vs diagonal TIC.

(d) :Generalization gap vs lower bound TIC.

Figure 11: Small-scale experiments: comparison of the TIC estimate and Generalization Gap.

Table 8: Correlation: TIC Estimates $\mathrm{Tr}(\boldsymbol{F}(\boldsymbol{\theta})^{-1}\boldsymbol{C}(\boldsymbol{\theta}))$ and Generalization Gap

| Dataset | Model | Spearman's | Kendall's $\tau$ | Pearson's Correlation |
|---|---|---|---|---|
| Tiny MNIST | 2-NN | 0.532 | 0.38 | 0.27 |
| Tiny MNIST | 3-NN | -0.62 | -0.427 | -0.679 |
| Tiny MNIST | 3-NN w/ SC | 0.258 | 0.175 | -0.332 |
| Tiny MNIST | 3-LNN | 0.292 | 0.271 | 0.305 |
| Tiny MNIST | 3-LNN w/ SC | 0.886 | 0.75 | 0.922 |

Table 9: Correlation: TIC Estimates $\mathrm{Tr}(\boldsymbol{F}_{\mathrm{blockdiag}}(\boldsymbol{\theta})^{-1}\boldsymbol{C}_{\mathrm{blockdiag}}(\boldsymbol{\theta}))$ and Generalization Gap

| Dataset | Model | Spearman's | Kendall's $\tau$ | Pearson's Correlation |
|---|---|---|---|---|
| Tiny MNIST | 2-NN | 0.524 | 0.372 | 0.288 |
| Tiny MNIST | 3-NN | -0.549 | -0.366 | -0.622 |
| Tiny MNIST | 3-NN w/ SC | 0.364 | 0.244 | -0.08 |
| Tiny MNIST | 3-LNN | 0.26 | 0.257 | 0.252 |
| Tiny MNIST | 3-LNN w/ SC | 0.937 | 0.823 | 0.944 |

Table 10: Correlation: TIC Estimates $\mathrm{Tr}(\boldsymbol{F}_{\mathrm{diag}}(\boldsymbol{\theta})^{-1}\boldsymbol{C}_{\mathrm{diag}}(\boldsymbol{\theta}))$ and Generalization Gap

| Dataset | Model | Spearman's | Kendall's $\tau$ | Pearson's Correlation |
|---|---|---|---|---|
| Tiny MNIST | 2-NN | -0.309 | -0.23 | -0.234 |
| Tiny MNIST | 3-NN | -0.297 | -0.203 | -0.415 |
| Tiny MNIST | 3-NN w/ SC | -0.176 | -0.128 | -0.415 |
| Tiny MNIST | 3-LNN | 0.275 | 0.237 | 0.288 |
| Tiny MNIST | 3-LNN w/ SC | 0.932 | 0.796 | 0.918 |

### D.3 ADDITIONAL RESULTS OF PRACTICAL-SCALE EXPERIMENTS

In this section, we present results of practical-scale setting that we have not presented in the main paper. We observed that in a small-scale setting, DNNs with a large number of layers and SC tend to have a high rank correlation in the TIC estimator and a strong correlation with the generalization gap. Since it is not computationally feasible to compute TIC in practical DNNs, this section details experimentalresults on the performance of two approximate estimators of TIC using diagonal approximation and its lower bound.

#### D.3.1 PRACTICAL-SCALE EXPERIMENTS: MNIST CASE

As shown in Figure 12, we observed the correlation between the TIC estimator and the generalization gap for the diagonal approximation and the lower bound. Both were found to be highly correlated and good estimators of the generalization gap.

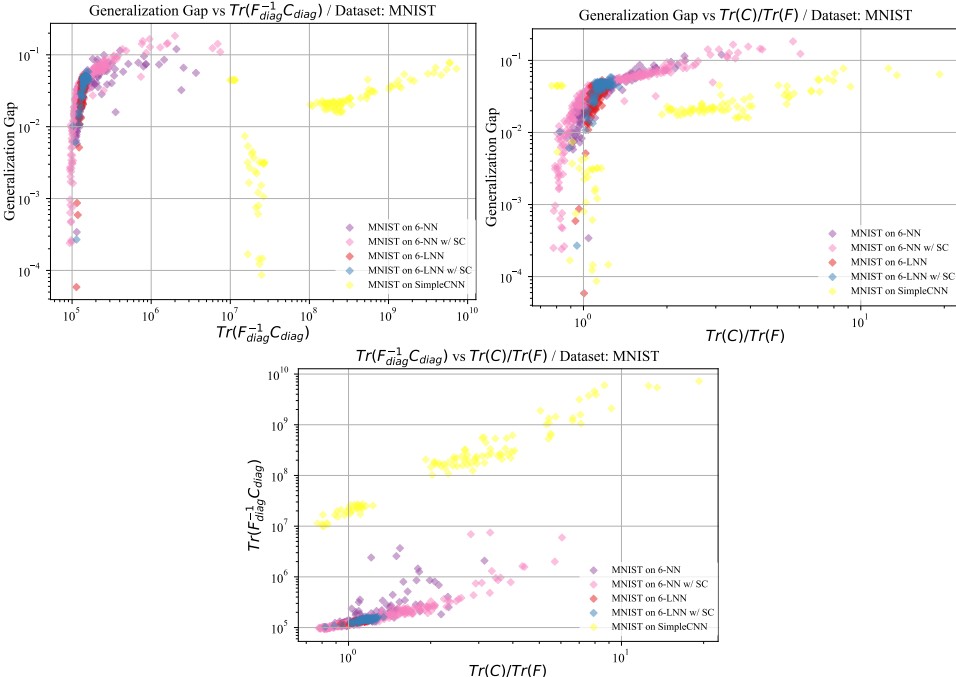

Figure 12: Practical-scale MNIST experiments: comparison of the TIC estimates.

Here, we also investigated whether the estimated TIC from lower bounds is due to only one of the components of the trace, respectively. As shown in figure 13, the value of trace itself was not correlated with the generalization gap. It was also confirmed that different models behaved differently. At the same time, we also confirmed that $\mathrm{Tr}\left(\boldsymbol{F}(\boldsymbol{\theta})\right)$ is a good approximation of trace $\mathrm{Tr}\left(\boldsymbol{H}(\boldsymbol{\theta})\right)$.

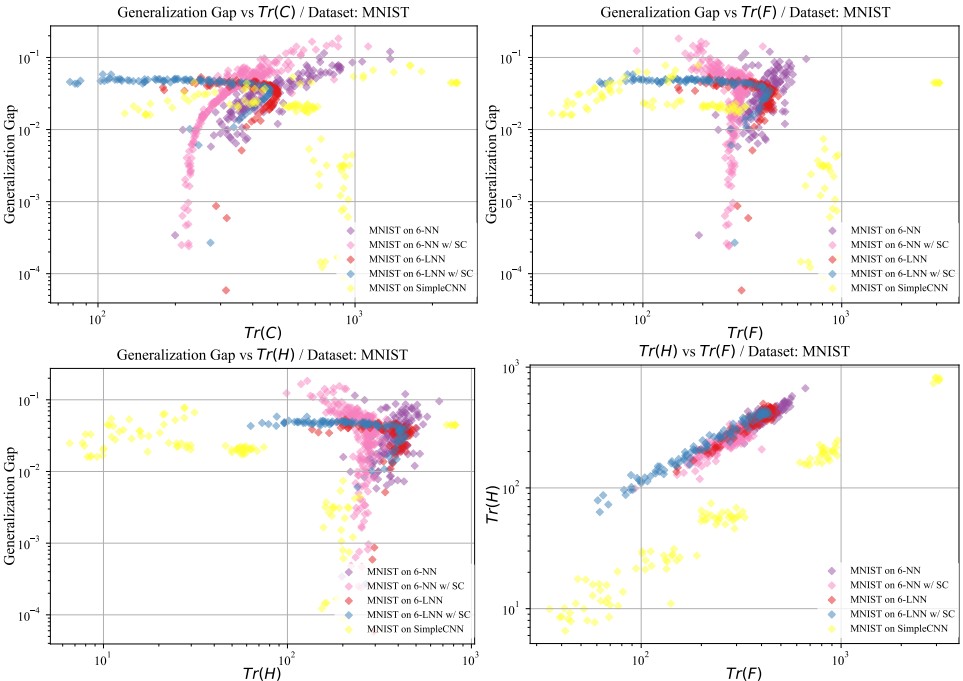

Figure 13: Practical-scale MNIST experiments: elements of the TIC estimates.

### D.3.2 PRACTICAL-SCALE EXPERIMENTS: CIFAR10 AND CIFAR100 CASE

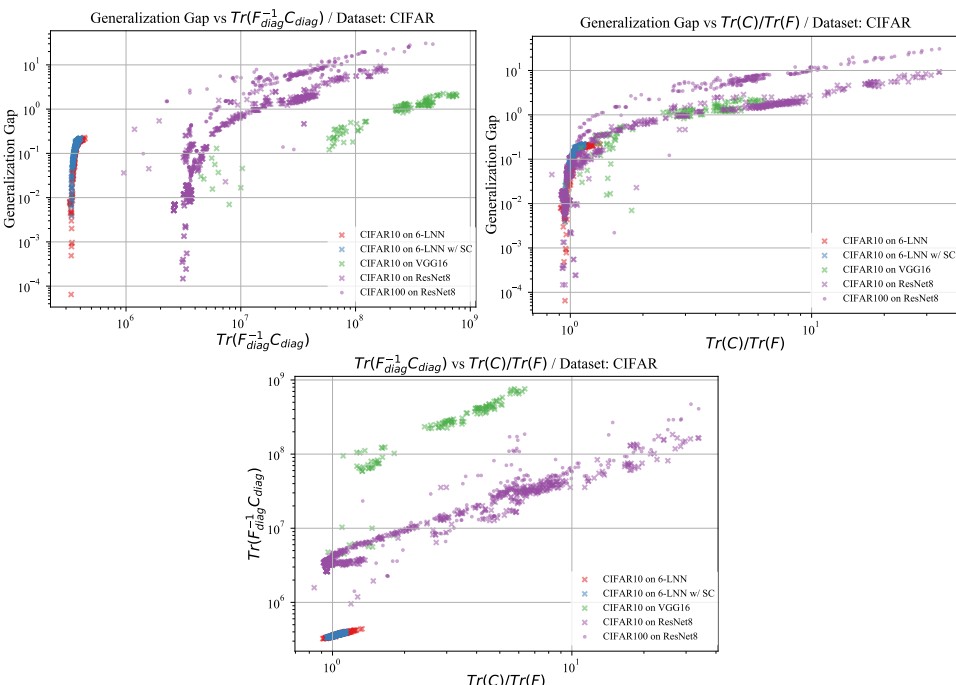

Figure 14: Practical-scale CIFAR experiments: comparison of the TIC estimates.

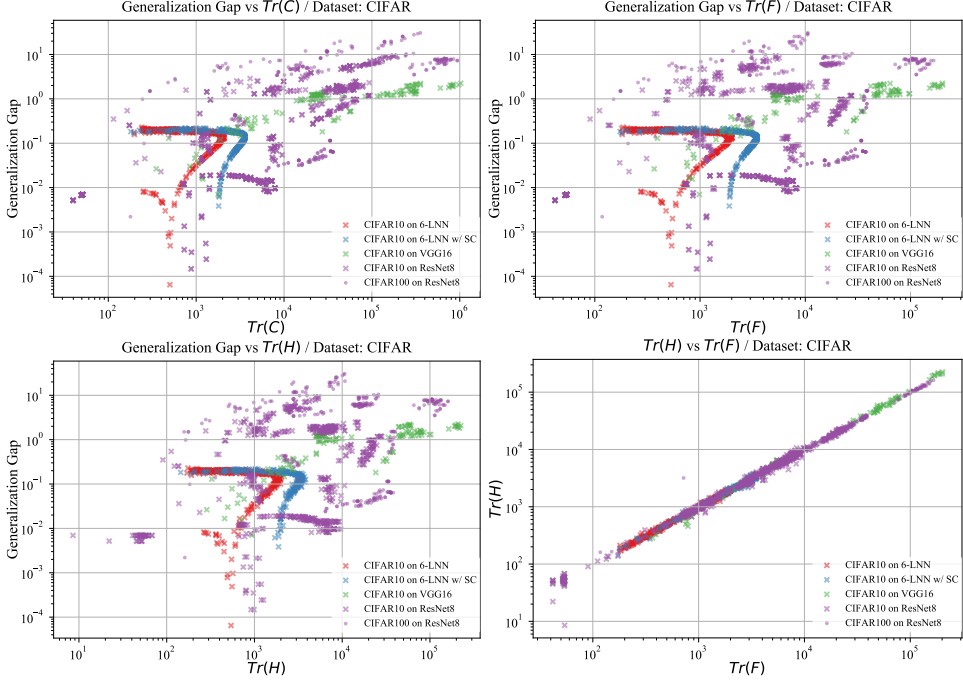

Figure 15: Practical-scale CIFAR experiments: elements of the TIC estimates.

### D.3.3 CORRELATION BETWEEN TIC ESTIMATES AND GENERALIZATION GAP IN TRAINING PROCESS

Within the scope of our experiments, we find that TIC can estimate the generalization gap even in the middle of learning for problem settings that are considered to belong to NTK.

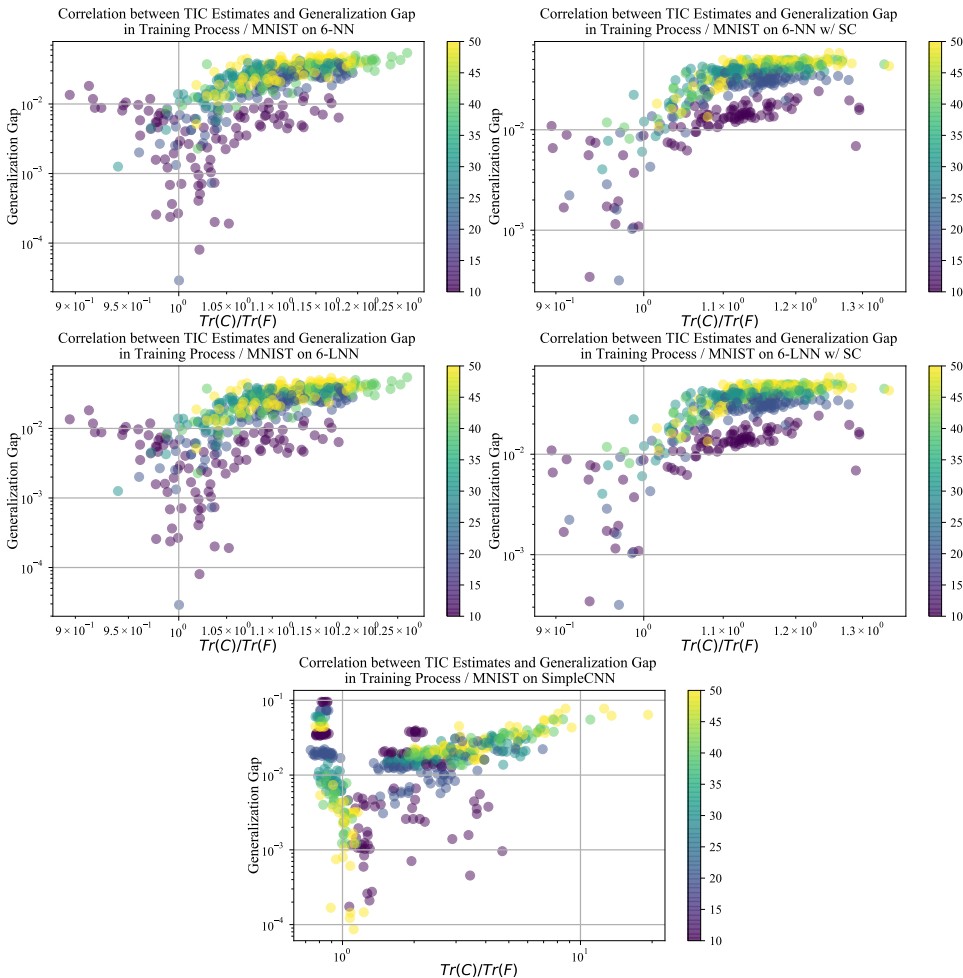

Figure 16: Correlation between generalization gap and TIC estimates in MNIST experiments, through training process. The color map shows the epoch.

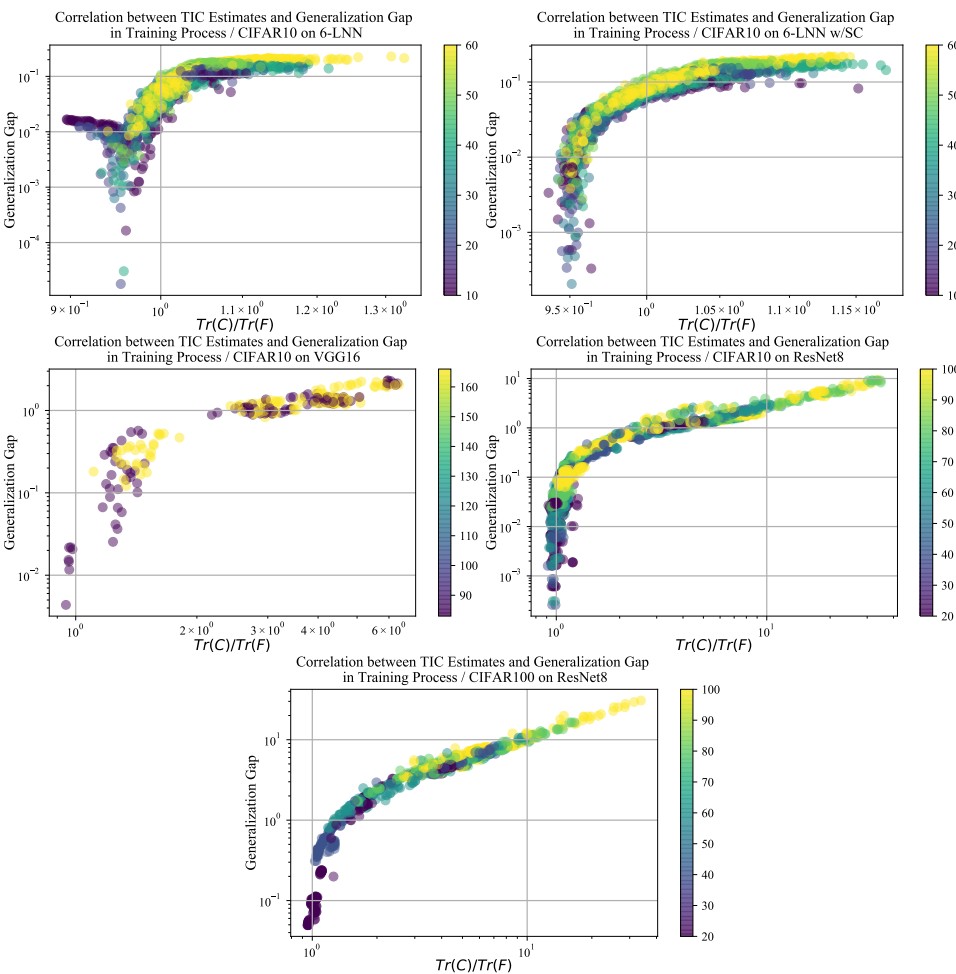

Figure 17: Correlation between generalization gap and TIC estimates in CIFAR experiments, through training process.The color map shows the epoch.

### D.4 CALCULATION TIME MEASUMENTS EXPERIMENT

First of all, we explain the execution time of the small NN/LNN case. In the case of using $\mathrm{Tr}\,(\boldsymbol{F}(\boldsymbol{\theta}))$ and $\mathrm{Tr}\,(\boldsymbol{C}(\boldsymbol{\theta}))$, the computation time by block diagonal approximation is only 40% faster than exact computation. The computational complexity should have been reduced from $O(d^3)$ to $O(d_l^3)$ by the block diagonal approximation. However, the overhead such as memory copying is dominant, and there is no significant difference in execution time compared to the theoretical amount of computation. From block diagonal to diagonal approximation, the experimental results show a reduction of up to 50% in computation time. The computational complexity is reduced from $O(d_l^3)$ to $O(d)$.

Secondly, the use of $\mathrm{Tr}\,(\boldsymbol{H}(\boldsymbol{\theta}))$ and $\mathrm{Tr}\,(\boldsymbol{C}(\boldsymbol{\theta}))$ requires up to 900% more time than the use of $\mathrm{Tr}\,(\boldsymbol{F}(\boldsymbol{\theta}))$ and $\mathrm{Tr}\,(\boldsymbol{C}(\boldsymbol{\theta}))$ in the Exact case. The choice to use $\mathrm{Tr}\,(\boldsymbol{F}(\boldsymbol{\theta}))$ instead of $\mathrm{Tr}\,(\boldsymbol{H}(\boldsymbol{\theta}))$ is therefore justified in terms of the reduction in computational time. In the case of block diagonalization, the speed-up was only a few percent. In the case of diagonalization, no significant speed-up was observed, but the speed-up was more than 40 times when Trace's approximation was performed using the Hutchinson method. However, the estimation using $\mathrm{Tr}\,(\boldsymbol{F}(\boldsymbol{\theta}))$ and $\mathrm{Tr}\,(\boldsymbol{C}(\boldsymbol{\theta}))$ is faster in the Small Scale problem setting.

It is observed that in such a small scale problem setting, there is a 50 times difference in execution time between using $\mathrm{Tr}\,(\boldsymbol{H}(\boldsymbol{\theta}))$ and $\mathrm{Tr}\,(\boldsymbol{C}(\boldsymbol{\theta}))$ with Exact TIC and computing $\mathrm{Tr}\,(\boldsymbol{F}(\boldsymbol{\theta}))$ and $\mathrm{Tr}\,(\boldsymbol{C}(\boldsymbol{\theta}))$ simultaneously and using diagonal approximation.

This speed-up should be more significant for larger models. As mentioned in section 2.2, it was not feasible to calculate of $\mathrm{Tr}\,(\boldsymbol{H}(\boldsymbol{\theta}))$ in ResNet-8 which requires more than 2,00 TB of memory without approximation. While execution time is also important, the most important aspect is that it is made possible to calculate TIC by approximation.

Secondly, we compare the execution time at practical scale. Among the cases where $\mathrm{Tr}\,(\boldsymbol{F}(\boldsymbol{\theta}))$ and $\mathrm{Tr}\,(\boldsymbol{C}(\boldsymbol{\theta}))$ is used, we investigate how much the time can be reduced when $\mathrm{Tr}\,(\boldsymbol{F}(\boldsymbol{\theta}))$ and $\mathrm{Tr}\,(\boldsymbol{C}(\boldsymbol{\theta}))$ are computed simultaneously compared to the case where they are computed separately. In the case of Small Scale and Practical Scale networks, the time is reduced by half, but in the case of SimpleCNN, VGG16, ResNet8, etc., the time is not reduced significantly. For relatively small models, which could be reduced by about 50%, the approximation by simultaneous $\mathrm{Tr}\,(\boldsymbol{F}(\boldsymbol{\theta}))$ and $\mathrm{Tr}\,(\boldsymbol{C}(\boldsymbol{\theta}))$ calculations was faster than using the Hutchinson method. In contrast. For networks with a large number of dimensions in the final layer, such as VGG16, etc., the calculation of $\mathrm{Tr}\,(\boldsymbol{C}(\boldsymbol{\theta}))$ and a fast approximation of $\mathrm{Tr}\,(\boldsymbol{H}(\boldsymbol{\theta}))$ resulted in a speedup of 10%-25% faster than the simultaneous calculation of $\mathrm{Tr}\,(\boldsymbol{F}(\boldsymbol{\theta}))$ and $\mathrm{Tr}\,(\boldsymbol{C}(\boldsymbol{\theta}))$.

Table 11: **Full Details of Runtime Measuments Experimemnt.** Unit: second

| Dataset | Model | Exact w/FC | Block Diag w/FC | Diag w/FC | Lower Bound w/FC | Exact w/HC | Block Diag w/HC | Diag w/HC | Lower Bound w/HC | Lower Bound w/HC by Hutchinson's Method |
|---|---|---|---|---|---|---|---|---|---|---|
| Tiny MNIST | 2-wide NN | 10.9093 | 6.7516 | 3.7273 | 1.9069 | 105.4282 | 96.0647 | 92.979 | 82.0081 | 2.7814 |
| Tiny MNIST | 3-NN | 3.8641 | 3.8435 | 3.7937 | 1.994 | 23.9083 | 22.4055 | 21.7705 | 22.4203 | 2.5705 |
| Tiny MNIST | 3-NN w/ SC | 3.8999 | 3.8568 | 3.8239 | 2.0225 | 26.2303 | 24.9561 | 25.0151 | 24.9217 | 2.8932 |
| Tiny MNIST | 3-LNN | 3.8656 | 3.847 | 3.7991 | 1.9521 | 23.188 | 22.2685 | 22.1641 | 21.6511 | 2.8645 |
| Tiny MNIST | 3-LNN w/ SC | 3.8771 | 3.8962 | 3.8159 | 1.9594 | 23.8571 | 22.8106 | 23.0338 | 22.0721 | 2.7198 |
| MNIST | 6-NN | N/A | N/A | 3.7235 | 1.9172 | N/A | N/A | N/A | N/A | 2.8949 |
| MNIST | 6-NN w/ SC | N/A | N/A | 3.6813 | 1.9459 | N/A | N/A | N/A | N/A | 3.3679 |
| MNIST | 6-LNN | N/A | N/A | 3.4822 | 1.9148 | N/A | N/A | N/A | N/A | 2.8089 |
| MNIST | 6-LNN w/ SC | N/A | N/A | 3.502 | 1.9329 | N/A | N/A | N/A | N/A | 2.9382 |
| MNIST | Simple CNN | N/A | N/A | 4.7059 | 2.6337 | N/A | N/A | N/A | N/A | 8.7958 |
| CIFAR10 | 6-LNN | N/A | N/A | 3.6742 | 1.9295 | N/A | N/A | N/A | N/A | 4.1043 |
| CIFAR10 | 6-LNN w/ SC | N/A | N/A | 3.7003 | 1.9453 | N/A | N/A | N/A | N/A | 2.8996 |
| CIFAR10 | ResNet8 | N/A | N/A | 12.4677 | 10.0031 | N/A | N/A | N/A | N/A | 90.9633 |
| CIFAR10 | VGG16 | N/A | N/A | 13.3077 | 11.3721 | N/A | N/A | N/A | N/A | 60.716 |
| CIFAR100 | ResNet8 | N/A | N/A | 12.4292 | 9.9655 | N/A | N/A | N/A | N/A | 88.7423 |

## D.5 Experiments with $d/n$ changes within TinyMNIST

We created a restricted dataset, SmallTinyMNIST, which uses only 5% of the TinyMNIST data.

Table 12: Additional problem settings are highlighted in bold text

| Category | Problem Setting: Dataset & Model | Ratio: $d/n$ |
|---|---|---|
| | TinyMNIST on 2-NN w/o SC | 0.09 |
| Small Scale | TinyMNIST on 3-NN w/o SC, 3-NN w/ SC | 0.02 |
| Data Size <1MB | TinyMNIST on 3-LNN w/o SC, 3-LNN w/ SC | 0.02 |
| Model <50KB | **SmallTinyMNIST on 3-LNN w/o SC** | **0.36** |
| | **TinyMNIST on Wide 3-NN w/ SC** | **13.90** |

We conducted training evaluations using SmallTinyMNIST to investigate the relationship between the TIC estimator and the generalization gap for patterns with relatively large $d/n$ ratios.

Furthermore, we prepared a network with a large number of $d$ as a 3-layer Wide-NN, and confirmed the effectiveness of TIC in settings where the $d/n$ ratio exceeds 10.

The distributions of the loss and the generalization gap are shown in figure 18. For the comparative purpose, problem settings with the same or relative networks are included in comparison plots and tables. The correlation between the TIC and the generalization gap is shown in figure 19.

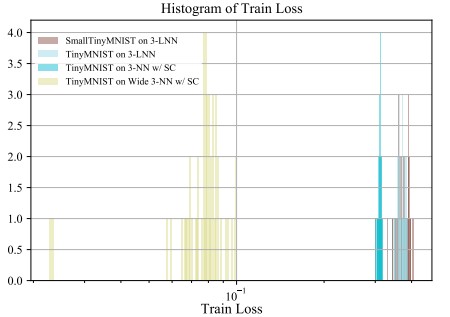 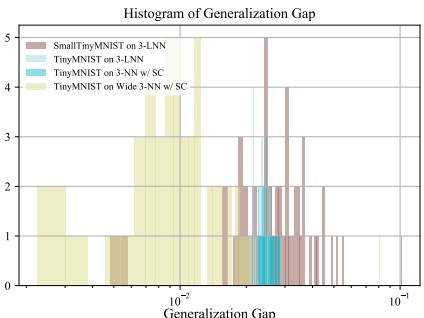

Figure 18: Distribution of training loss and generalization gap on the trained models

Table 13: Correlation: TIC estimates $\mathrm{Tr}(\boldsymbol{C}(\boldsymbol{\theta}))/\mathrm{Tr}(\boldsymbol{F}(\boldsymbol{\theta}))$ and generalization gap

| Model | Dataset | Spearman's | Kendall's $\tau$ | Pearson's Correlation |
|---|---|---|---|---|
| 3-LNN | Tiny MNIST | 0.277 | 0.238 | 0.256 |
| 3-LNN | Small Tiny MNIST | 0.806 | 0.622 | 0.811 |
| 3-NN w/ SC | Tiny MNIST | -0.19 | -0.137 | -0.347 |
| 3-Wide NN w/ SC | Tiny MNIST | 0.834 | 0.656 | 0.967 |

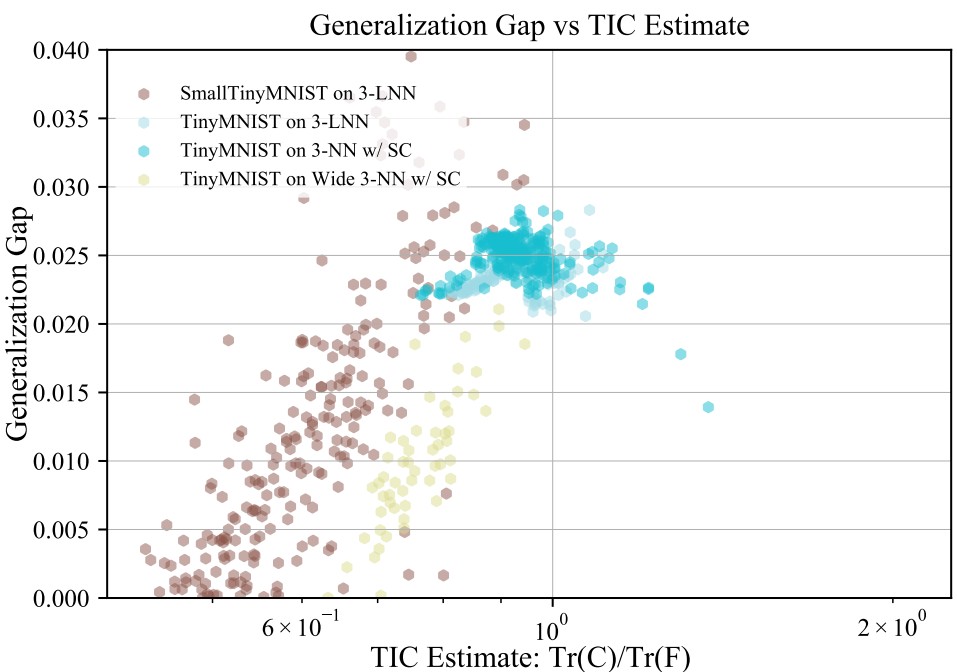

Figure 19: Relationship between TIC estimates and generalization gap

