# OpenReview forum: "Takeuchi's Information Criteria as Generalization Measures for DNNs Close to NTK Regime"
_ICLR.cc/2022/Conference — ICLR 2022 Submitted_

### Official Review · Reviewer_fZpy · 2021-10-27

**Correctness:** 3
**Technical Novelty And Significance:** 3
**Empirical Novelty And Significance:** 3
**Recommendation:** 5
**Confidence:** 3

**Details Of Ethics Concerns:**

No.

**Main Review:**


Contributions:

1. The authors propose to use a simple but effective metric, TIC, to estimate the generalization gap.

2. The authors show how to speed up the estimation via Monte-Carlo sampling.

3. The experiments validate the utility of TIC by showing a large correlation between TIC and the generalization gap.

Concerns:

1. The authors claim that: “estimated TIC well correlates generalization gaps under the conditions that are close to NTK regimes. Outside the NTK regime, such correlation disappears.” Do the authors mean a large d/n when referring to NTK regimes? What about the other requirements in NTK, for example, small initialization.

2. In Figure4 and Figure3, how do the authors calculate the Pearson’s coefficient (for example, by repeated experiments?)? I have not found the descriptions.

3. Beyond the generalization gap, I am wondering how large the test error is. Are the models trained to be zero training loss?

4. I think the authors can conduct some experiments (maybe small-scale is enough) on TinyMNIST with a large d/n, does the Pearson’s coefficient act as MNIST?

Some other comments:

1. Could the authors explain further what is “model distribution” q_{\theta}? Did the authors consider a Bayesian regime where the model distribution represents the distribution of the parameters? I am still puzzled about the differences between C(\theta) and F(\theta).

2. What is the role of G(\theta) in the main text? Do the authors use it during the TIC estimation process?

Typos:
1. Page3: they a different different distribution.

2. The font size of Fig2 is too small.

3. d in d/n (experiment section): d is the number of parameters?

**Summary Of The Paper:**

This paper focuses on a fascinating, challenging problem: evaluating the generalization performances of neural networks based on the training set with some criteria. Specifically, the authors propose to measure the generalization ability via Takeuchi’s Information Criteria (TIC). Besides, the authors approximate the TIC to accelerate the process of estimation. The proposed method is promising and works well when d/n is large (where d = #parameters and n = #samples).
However, I am still puzzled about some details. I will raise my score if the authors could clarify those concerns (see the following).

**Summary Of The Review:**

Overall, I think this paper focuses on a challenging topic and has the potential to be accepted. I would like to see more explanations on the details and update my score after the rebuttal period.


==========================
After reading the authors' response, I decided to keep my scores unchanged.

---

> ### Author Response · Authors · 2021-11-23
> **Official Response to Reviewer fZpy (1/2)**
>
> Thank you for taking your time to review our manuscript.
> Below we address specific concerns and comments:
>
> ### Concerns
>
> > Do the authors mean a large d/n when referring to NTK regimes? What about the other requirements in NTK, for example, small initialization?
>
> For the case of Fully-connected DNNs, as you asked, it requires initializing the weights to follow random Gaussian distribution.
> For the width m we do a scale transformation of $O(\sqrt{m})$.
> In addition, the number of parameters $d$ in each layer should be large enough for the training data $n$ and need to normalize the input.
>
> > In Figure4 and Figure3, how do the authors calculate the Pearson’s coefficient (for example, by repeated experiments?)? I have not found the descriptions.
>
> In the hyperparameter search range, we first prepare a checkpoint of models trained by different hyperparameters. We then calculate the pairs of generalization gaps of these models and the TIC estimates. As these pairs exist for a combination of hyperparameters, we use them to calculate the respective correlation coefficients. The Pearson's correlation coefficient is calculated as the covariance between the value of the generalization gap and the estimate of the TIC estimates divided by the product of the standard deviation of the generalization gap and the standard deviation of the estimate of the TIC estimates.
>
> > Beyond the generalization gap, I am wondering how large the test error is. Are the models trained to be zero training loss?
>
> Thank you for pointing this out. In some problem settings, the loss is quite close to 0, but in some experimental settings, it is not (e.g., the 2-layer NN seemed to be difficult to train).
> In our revision, we have added a histogram of the loss and generalization gap for each experimental setup in the Appendix C.3.
>
> > I think the authors can conduct some experiments (maybe small-scale is enough) on TinyMNIST with a large $d/n$, does the Pearson’s coefficient act as MNIST?
>
> We have added two new experimental settings to investigate this problem. The current small scale uses only TinyMNIST and with only a small $d/n$ problem setting.
> So, while using TinyMNIST, we experimented with two approaches to increase $d/n$.
> 1.  Increase the model size ($d \uparrow$)
> 2.  Limit the number of data  ($n \downarrow$)
>
> The results show that the correlation coefficient between the TIC estimate and the generalization gap approaches $1$ for large values of $d/n$, even when TinyMNIST is used. Details of the experimental results are also added in the Appendix D.5 of our revision.

---

> ### Author Response · Authors · 2021-11-23
> **Official Response to Reviewer fZpy (2/2)**
>
> ### Comments
>
> > Could the authors explain further what is “model distribution” q_{\theta}? Did the authors consider a Bayesian regime where the model distribution represents the distribution of the parameters? I am still puzzled about the differences between C(\theta) and F(\theta).
>
> Thank you for pointing this out. We clarify this part.
> In particular, Frederik et al. [1] has pointed out that the definition of $F$ is ambiguous, depending on the research area and community.
> The correct definition of $F$  in the field of deep learning these days is as follows.
>
> $F=\sum^{\text {data}} \sum^{\text {class}} q \left(\frac{\partial l}{\partial \theta}\right)\left(\frac{\partial l}{\partial \theta}\right)^{T}$
>
> where $l=-\log p$ as negative log-likelihood per class per data sample,  $q$ is out put of DNN model and $p$ is the target label.
>
> Here we have written the expectation in the form of a summation.
> On the other hand, the definition of $C$  is as follows.
>
> $C=\sum^{\text {data}} \sum^{\text {class}} p \left(\frac{\partial l}{\partial \theta}\right)\left(\frac{\partial l}{\partial \theta}\right)^{T}$
>
> Thus, the model distribution $q_{\theta}$ represents the distribution of the output of the DNN model $\theta$.
> Since $C$ relies on the data distribution, the class summation is fast in a one-hot label problem setting as it only extracts one element in effect.
>
> > What is the role of G(\theta) in the main text? Do the authors use it during the TIC estimation process?
>
> Assuming the operation of transforming the output of the model in the final layer to be considered as probability, as in $q = sigmoid(z)$, the definition of $G$ is as follows:
>
> $G=\sum^{\text {data}} \sum^{\text {class}} \left(\frac{\partial z}{\partial \theta}\right)\left(\frac{\partial^{2} l}{\partial z^{2}}\right)\left(\frac{\partial z}{\partial \theta}\right)^{T}$
>
> $G$ is called the Generalized Gauss-Newton matrix and is an approximation of $H$.
> In general deep learning settings (Assumption 3.1), J Martens [2] has proven that $H$ equals $F$ through $G$.
> By using this trick, we justify the replacement of the calculation of $H$ by the calculation of $F$.
> In the calculation of $H$ in our experiments, we have calculated $G$.
>
> [1] Kunstner, Frederik, Philipp Hennig, and Lukas Balles. "Limitations of the empirical fisher approximation for natural gradient descent." Advances in neural information processing systems 32 (2019): 4156-4167.
> [2] Martens, James. "New Insights and Perspectives on the Natural Gradient Method." Journal of Machine Learning Research 21 (2020): 1-76.

---

### Official Review · Reviewer_Z8n4 · 2021-11-01

**Correctness:** 2
**Technical Novelty And Significance:** 2
**Empirical Novelty And Significance:** 2
**Recommendation:** 3
**Confidence:** 3

**Main Review:**

One positive point here is that the authors have really run a very large amount of experiments, which deserves credit. On the other hand, the paper is extremely poorly written and organised to the point where it just doesn't look like a serious submission by the standards of a top tier conference such as ICLR. The text is rambling, the theoretical "results" can only be understood by reading the related works rather than by reading the current paper. Assumptions and statements are often vague, not only in content but even in pagination: there are "propositions" which merge into the text with no clear beginning or end. So called "Proofs" also appear to be more like an informal and aimless discussion that ends with a reference to the supplementary, where one finds another casual and rambling discussion which ends in a reference to related works. Even the experiments are not discussed in an organised way and many of the graphs are blurry with illegible legends. The english is poor (this wouldn't be a problem if the paper was well organised). Furthermore, other ability-agnostic metrics attest to the simple lack of time/effort invested by the authors into the polishing: there are issues with punctuation/presentation and some of the key references are incorrectly cited.

Last but not least, there is (pretty much by the authors' own admission) almost no original content in the proposed methods (except approximations (2) and (3) from the summary above).






=======================After Rebuttal==================


I took a quick look at the rebuttal. The authors seem to commit to making many of the necessary changes that I asked for (which I am happy about), but some of the replies are still relatively vague. Assuming the current pdf is the last revision, it is difficult to compare to earlier versions (it would be customary to highlight the changes in a specific colour for instance). At least some of the problems I mentioned are still there, including the inaccurate big picture description of the related works.


Still, I think the paper is below the borderline for ICLR including assuming every issue was fixed (which is not the case). I would change my score to 4 if it were possible but the only allowed scores are 3 and 5 and I still think 5 is a bit too much for the paper in its current state.




/
===========
Specific comments/questions
===========

1.  The motivation behind the main method, and the reason why the term $\tr(H^{-1}C)$ appears, is not provided in the paper. I understand that this is "known" from the related work, but it would be nice to make the paper more accessible by explaining it. Furthermore, it is claimed in the main paper (page 4 near equation (3)) that "a more detailed proof is provided in Appendix A1". This is not true. First of all, Appendix A1 is quite long and it is not immediately clear which part of it they are referring to. After some thought it appears to be the end of Appendix A.1.2 on page 14, specifically equations (20) to (24) inclusive. It is not explained why 22 and 23 converge to $\frac{1}{2}\tr(H^{-1}D)$. Note that there is also an extra factor of $n$ in the definition of $b$ in the appendix compared to in the main text.

2. In the introduction, the TIC is introduced via formula (1) and the matrices H,C at this point were only defined as "Hessian" and "Covariance" respectively, which is not enough information even for an informal reading. At least a pointer to the more formal definition on page 3 (equation (2)) would be nice.

3. Question: on page 3 (near the bottom), you say that $H(\theta)$ is a measure which doesn't depend on the distribution of the input data. What do you mean? As far as I understand, this quantity still involves an expectation over a sample, which depends on the sampling distribution.


4. One big problem with this paper is that it is not clearly explained why the method provided is specific to the NTK regime and assumptions and statements are not rigorously stated. For instant, Assumption 2.1 is not clear. First of all, $q_\theta$ is not defined anywhere in the paper as far as I can tell. Secondly, it appears the authors are stating that there can only exist one set of parameters that minimizes the loss. This is certainly not true *at face value* for neural networks. I agree that it is approximately true for the NTK itself, as the authors clumsily explain in appendix A1.1. Still, it is no longer clear if we are speaking about an "assumption" or something which is deduced from that part of the appendix. The "uniqueness" is kind of trivial once we accept that we define the "solution" as whatever happens when we run gradient descent to convergence (this isn't even specific to NNs in the NTK regime). The second assumption of asymptotic normality is not explained in detail.

5. Question: what is going on in the graph at the bottom of Figure 3. It appears inconclusive (same for the graphs on page 19)

6. What is the strategy employed to use the TIC to improve hyper parameter search. As far as I can tell, the only sentence in the relevant section (Section 5) which attempts to explain that is the following: " To achieve stable optimization in SHA, we apply the TIC values for the intermediate loss. There is no formula anywhere in the section which involves such an "intermediate loss", or any reasonable explanation of the optimization strategy there.


7. Question: by "Linear neural networks" do the authors really mean neural networks without any activation functions? It would seem that large scale experiments are difficult to perform in this context as linear functions cannot match the ground truth well at all in realistic scenarii


8. The section on related works is a bit chaotic and incomplete. The authors are splitting the work into two categories: one the one hand "generalisation bounds" (with one citation, which deals with a PAC Bayesian approach) and on the other hand "complexity measures", with a rather incomplete description of some of the work in the statistical learning theory of neural networks. The logic in the three first lines of page three doesn't seem to stand: the strong negative correlation between the spectral norm of the layers and generalization is a point in favour of the validity of the approach, not against, as appears to be intimated by the authors.

9. The authors seem to hint that the reason one cannot use the AIC is because of the misspecification of the model, but I don't think that is true. There are plenty of reasons in the SLT of NNs why parameter counting is inadequate to explain generalization (including in the realisable, "well specified" case), I think these reasons still hold here: the optimization of a NN in NTK regime behaves more like kernel ridge regression than like the problem of finding the global optimum of the loss minimization problem over the whole set of parameters.


10. The concept of "singular models" is not explained in the introduction, though it is referenced many times. The explanation which is given later (page 13, top of Section A.1.2. in the supplementary!) is not very satisfactory either.

11.Question I might have missed that but what is "SC" in the experiments when describing the various architectures?


12. Question for the authors (I might be wrong here, I am just genuinely curious): on page 5 in the explanation of the block diagonalization procedure, you claim that this approach reduces the complexity from $O(d^3)$ to $O(d)$. That is a bit confusing....
 If the width is $d_1$, I guess the complexity should be $d_1^3$ to invert each individual matrices, times $d/d_1$ (the number of layers), which gives $d_1^2 d$.

Are you surreptitiously treating the width of the intermediary layers as constant?!?



 /
=====================
Minor issues, typos, etc. (Highly non exhaustive)
=====================

1. abstract: "In fact, theory indicates..." =======> "In fact, the theory indicates..."
2. Beginning of the last two paragraphs of page 1: determinant ("the") missing before "generalization gap"
3. Top of page 2: "$\hat{\theta}$ is the solution to the empirical loss" ===>
"$\hat{\theta}$ minimizes the empirical loss"
4. Page 2, "We provide empirical and theoretical evidences" =====> "We provide empirical and theoretical evidence"
(uncountable)
5. page 2 "estimate TIC with less computations" ====>"estimate TIC with fewer computations"
6. Page 5, assumption B2: improve sentence structure
7. Middle of page 5: "In NTK's regime" ====> "In the NTK regime"
 8. Page 5 "for estimate $\tr(H)$ fast"====>  "to estimate $\tr(H)$ fast" OR  "for estimating $\tr(H)$ fast"
9. Page 7 (question)  "we can confirm that the value and the rank correlation are kept": what does this mean?
10. Page 8, section 4.4, third line, there is an extra space after $F(\theta)$.
11. Page 9: "Successive halving algorithm shows promising..." =====>"The successive halving algorithm shows promising..."

12. References [2] and [1] appear incorrectly cited. [2] is published in JMLR and [1] is published in AiStats.

13. Conclusion: "even if the model is not completed to train": what does this mean?
14. Page 15 above equation (25):  "the matrix H and D" ====> "the matrices H and D"
15. Below equation (25): "respective matrice" ===> "respective matrices.
16. Page 16:  "when publication ===>"when published" OR "after publication"
17. Page 16 "For what hyperparameter in what range to search, we follow configuration of..." ====>"To set the range in which to search for each hyperparameter, we follow the configuration of..."
18. Page 29: "Exact case" is capitalised in the middle of a sentence. As is "small Scale".


19. The last two sentences of page 12 (at the beginning of the Appendix) mean the same thing. I guess one of them should be deleted.








/
==========
References
==========

[1] Valentin Thomas, Fabian Pedregosa, Bart van Merrie ̈nboer, Pierre-Antoine Mangazol, Yoshua Ben- gio, and NL Roux. Information matrices and generalization. 2019 arXiv preprint

(Appears to have in fact been published at AiStats 2020  with the title " On the interplay between noise and curvature and its effect on optimization and generalization". )


[2] James Martens. New insights and perspectives on the natural gradient method. JMLR 2020


[3] Haim Avron and Sivan Toledo. Randomized algorithms for estimating the trace of an implicit sym- metric positive semi-definite matrix. Journal of the ACM 2011



**Summary Of The Paper:**

The authors experimentally investigate the ability of Takeuchi's information criterion to predict generalization gaps in various neural network architectures, with a particular focus on the Neural Tangent Kernel (NTK) regime. This was previously done only on small scale networks [1], and the authors extend it to state-of-the-art large scale architectures. In order to do this successfully, the authors propose to use various approximations, including the following:
(1) The equivalent representation of the expected Hessian matrix H via the Gauss-Newton Matrix  (with the equivalence proven in [2]).
(2) A block diagonalization approach that ignores correlations between layers.
(3) An even coarser approximation that consists in ignoring the off-diagonal elements of the relevant matrices H (expected hessian) and C (correlaction of the gradient)
(4) Hutchinson's method to estimate the trace of the Hessian (from [3])


The authors run a vast number of experiments and demonstrate that Takeuchi's information criterion exhibits significant correlation with the generalization gap and that many of their approximations perform well. The authors also propose to use the TIC to improve hyperparmeter search, with encouraging results.





/
==========
References
==========

[1] Valentin Thomas, Fabian Pedregosa, Bart van Merrie ̈nboer, Pierre-Antoine Mangazol, Yoshua Ben- gio, and NL Roux. Information matrices and generalization. 2019 arXiv preprint

(Appears to have in fact been published at AiStats 2020  with the title " On the interplay between noise and curvature and its effect on optimization and generalization". )


[2] James Martens. New insights and perspectives on the natural gradient method. JMLR 2020


[3] Haim Avron and Sivan Toledo. Randomized algorithms for estimating the trace of an implicit sym- metric positive semi-definite matrix. Journal of the ACM 2011


**Summary Of The Review:**

The authors deserve credit for funning a lot of experiments. However, there is minimal original contribution. Furthermore, the paper is poorly written and organised, and of insufficient mathematical rigour to warrant publication at ICLR.

---

> ### Author Response · Authors · 2021-11-23
> **Official Response to Reviewer Z8n4 (1/2)**
>
> Thank you for taking your time to review our manuscript. Below we address specific comments and questions:
>
>  > The motivation behind the main method, and the reason why the term $tr(H^{-1}C)$appears, is not provided in the paper. I understand that this is "known" from the related work, but it would be nice to make the paper more accessible by explaining it. Furthermore, it is claimed in the main paper (page 4 near equation (3)) that "a more detailed proof is provided in Appendix A1". This is not true. First of all, Appendix A1 is quite long and it is not immediately clear which part of it they are referring to. After some thought it appears to be the end of Appendix A.1.2 on page 14, specifically equations (20) to (24) inclusive. It is not explained why 22 and 23 converge to  $\frac{1}{2}tr(H^{-1}C)$
>
> and
>
> > In the introduction, the TIC is introduced via formula (1) and the matrices H,C at this point were only defined as "Hessian" and "Covariance" respectively, which is not enough information even for an informal reading. At least a pointer to the more formal definition on page 3 (equation (2)) would be nice.
>
> We are sorry that this part was not clear in the original manuscript. In our revision, we made a correction regarding references to appendixes. We modified the footprint to refer to the definition of H and C where these terms appear at the first time.
> With respect to equation 23, since taking the expectation of the left-hand side by $p$ coincides with the right-hand side, as a result, they cancel each other out and become zero.
> Equation 22 can be evaluated by using the asymptotic normality and the expectation of the quadratic form.
> Details have been amended in the form of citations to previous studies.
> This clarification is important and is the issue we would like to address.
>
> > Note that there is also an extra factor of  $n$ in the definition of $b$  in the appendix compared to in the main text.
>
> Thank you for pointing out it. There was a wrong notation in the Appendix, which has been corrected.
>
> > Question: on page 3 (near the bottom), you say that $H$ is a measure which doesn't depend on the distribution of the input data. What do you mean? As far as I understand, this quantity still involves an expectation over a sample, which depends on the sampling distribution.
>
> We agree that the current description, especially for the motivation of considering H, is quite unclear.
> $H$ plays the role of sensitivity, such as perturbation of the input, and as you say, depends on the input.
> We will definitely fix it in the near future.
>
> > One big problem with this paper is that it is not clearly explained why the method provided is specific to the NTK regime and assumptions and statements are not rigorously stated. For instant, Assumption 2.1 is not clear. First of all,
> $q_\theta$ is not defined anywhere in the paper as far as I can tell. Secondly, it appears the authors are stating that there can only exist one set of parameters that minimizes the loss. This is certainly not true at face value for neural networks. I agree that it is approximately true for the NTK itself, as the authors clumsily explain in appendix A1.1. Still, it is no longer clear if we are speaking about an "assumption" or something which is deduced from that part of the appendix. The "uniqueness" is kind of trivial once we accept that we define the "solution" as whatever happens when we run gradient descent to convergence (this isn't even specific to NNs in the NTK regime).
>
> Thank you for pointing out the important point. We will reconstruct the paper overall to clarify the main point of this study.
> We had defined $q_\theta$ in the second paragraph of chapter 2.2, but it seemed to be in a difficult position to find.
> We intended to state that the TIC can be derived assuming the NTK regime, but we have many improvements to make. We assume that if the solution is in the framework of the NTK regime, the optimal solution is unique.
>
> > The second assumption of asymptotic normality is not explained in detail.
>
> Thank you for pointing out the most defect points of this paper.
> In our revision, we have modified the Appendix 1.2 and cited (White, 1982).
> As most challenging steps, in the NTK regime mentioned above, we need to consider, in particular, the central limit theorem in infinite dimensions and, strictly speaking, regularization.
> We will definitely try it in the near future.
>
> > Question: what is going on in the graph at the bottom of Figure 3. It appears inconclusive (same for the graphs on page 19)
>
> The correlation coefficient is small, which means that the TIC is not valid, which happens in regimes far away from NTK.
> We are arguing that TIC is effective near the NTK regime when d/n is taken as a positional measure of proximity to the NTK regime.

---

> ### Author Response · Authors · 2021-11-23
> **Official Response to Reviewer Z8n4 (2/2)**
>
> > What is the strategy employed to use the TIC to improve hyper parameter search. As far as I can tell, the only sentence in the relevant section (Section 5) which attempts to explain that is the following: " To achieve stable optimization in SHA, we apply the TIC values for the intermediate loss. There is no formula anywhere in the section which involves such an "intermediate loss", or any reasonable explanation of the optimization strategy there.
>
> Thank you for the helpful comment.
> Here, we focus only on the pruning of the trial, not on the general hyperparameter search.
> In a typical HPO trial pruning, the validation loss is evaluated during the training process, and the error in the estimator is $O(1/ \sqrt{n})$ for the number of data $n$. TIC reduces this error to $O(1/{n})$, so it is more efficient. As TIC can reduce this estimation error, we can expect to make more correct decisions.
> The term "intermediate loss" was written in the sense of validation loss evaluated during the training, but it was inappropriate and corrected.
>
> > Question: by "Linear neural networks" do the authors really mean neural networks without any activation functions? It would seem that large scale experiments are difficult to perform in this context as linear functions cannot match the ground truth well at all in realistic scenario
>
> Yes, we agree with you, and Figure 7 in the Appendix shows the histogram of train loss for each problem settings.
> It motivated us to experiment with nonlinear NNs, ResNet, and VGG, as LNNs do not perform well on practical scales.
>
> > The section on related works is a bit chaotic and incomplete. The authors are splitting the work into two categories: one the one hand "generalisation bounds" (with one citation, which deals with a PAC Bayesian approach) and on the other hand "complexity measures", with a rather incomplete description of some of the work in the statistical learning theory of neural networks. The logic in the three first lines of page three doesn't seem to stand: the strong negative correlation between the spectral norm of the layers and generalization is a point in favour of the validity of the approach, not against, as appears to be intimated by the authors.
>
> Thank you for pointing out it.
> We will reconstruct the paper overall to clarify the main point of this study.
>
> > The authors seem to hint that the reason one cannot use the AIC is because of the misspecification of the model, but I don't think that is true. There are plenty of reasons in the SLT of NNs why parameter counting is inadequate to explain generalization (including in the realisable, "well specified" case), I think these reasons still hold here: the optimization of a NN in NTK regime behaves more like kernel ridge regression than like the problem of finding the global optimum of the loss minimization problem over the whole set of parameters.
>
> Thank you for the helpful suggestion.
> We have discussed the general differences between TIC and AIC in terms of allowing for misspecified situations, but we consider that analysis from the perspective of kernel ridge regression and the like may provide a hint as to whether TIC is empirically valid even in regions considered to be outside the NTK Regime.
>
> > The concept of "singular models" is not explained in the introduction, though it is referenced many times. The explanation which is given later (page 13, top of Section A.1.2. in the supplementary!) is not very satisfactory either.
>
> We have corrected this point in the revised version.
>
> > Question I might have missed that but what is "SC" in the experiments when describing the various architectures?
>
> In fact, we described it in a footnote, but it seems to have been difficult to look for.
> SC stands for Skip Connection.
>
> > Question for the authors (I might be wrong here, I am just genuinely curious): on page 5 in the explanation of the block diagonalization procedure....?
>
> We are sorry that this part was not clear in the original manuscript.
> We implicitly set $d_{l}$ to be the number of parameters for the layer with the largest width.
>
>
>
> We appreciate that you found a lot of points that need to be improved in terms of minor issues, typos, etc.

---

### Official Review · Reviewer_2vba · 2021-11-02

**Correctness:** 3
**Technical Novelty And Significance:** 2
**Empirical Novelty And Significance:** 2
**Recommendation:** 3
**Confidence:** 3

**Main Review:**

Strength:
- Studying what causes a smaller generalization gap is an important problem in deep learning. Given that the NTK regime leads to zero training loss, the generalization gap directly leads to generalization performance in the NTK regime.
- Extensive empirical evaluation is important given the complexity of the deep learning model/dataset/training. The paper is working towards such a goal.

Weakness

- Main idea of TIC’s usefulness as a generalization measure is already demonstrated in Thomas et al., 2019. In comparison to Thomas et al., 2019, the authors’ claimed difference is investigation beyond small scale and studying DNNs of practical size which requires estimation of TIC. However, the general lesson of TIC as a generalization measure and using efficient approximation of TIC  with Tr(F^{-1}C) and Tr(C) / Tr(F) has been explored in (Thomas et al., 2019). Possibly a new contribution is the message about “close to NTK regime” and looking at the correlation/ranking coefficients as a function of overparameterization ratio (d/n).

- To estimate TIC, we require a separate data (validation)  as well as trained parameter \theta^*for H and C computation. In this case,  why not just compute validation acc/loss which we know is a good estimate of generalization performance? [know without training? But needs trained params…]

- The reviewer tried looking for detailed model descriptions used (beyond simple names) for experiments but did not appear in paper nor in code.

- Overall, there’s issues with clarity. The paper needs significant proofreading to the point that deters understanding the main message.
   - Numerous typos or un-proofread sentences throughout the paper
   - Some of the main figures are not fully legible or explained.  e.g. Figure 1/2 is not so legible due to small fonts. Figure 1 appearing below the footnote is confusing. Moreover, neither captions for Figure 1 / 2 are self-explanatory nor they are properly referenced in the main text. In Figure 4, what does the color bar indicate? (In the appendix some figures denote that color bars are epochs. Main figure should explain this as well)
    - Please put proper references to which figure/table the main text is explaining.

- HPO experiments are promising. However, just showing one set of cases in Figure 5, it’s unclear whether the proposed method and described phenomena is robust.

Questions

- Could you provide the argument / reference showing that layer-wise block diagonal structure of Hessian in the NTK regime (section 3.2 part 2)? As far as the reviewer is aware, there’s block diagonal structure for NTK regards to output unit or K-FAC approximation (irrespective of NTK limit) but is not familiar with layer-wise block diagonality.

- According to remark 2.3., one would consider WAIC/GIC should be a more principled information criteria to be used for models near the NTK regime. TIC is also intractable due to high computational cost. Shouldn’t one also try to approximate the WAIC as similarly be done in the paper? Is there a chance that various approximations taken for TIC make it more like WAIC?

Corrections:
- Remark 2.3: I believe the exact criteria of being in the NTK regime is not precisely defined but beyond linear models, one often needs the number of model parameters to be much larger that number of the data points.

Nit:
- Typo: p2 in contribution part two, “with totally 12 architectures”?
- Type: p3 “but they a different different distribution when computing the expectation”
- “n” is not defined early enough also often used as “p” (e.g. section 4.3).
- Term GIC (possibly generalized information criteria) is not defined or referenced.
- Typo: Section D.4: “As mentioned in section X” (there’s no section X), “2,00TB”

-------------------------------------------------------
Post rebuttal:
I thank the authors for the response and clarification. I have read the response and other reviews and my assessment still remains as is.

In regards to block-diagonal approximation in Karakida-Kazuki NeurIPS 2020, to my understanding the approximation(block-diagonal, block tri-diagonal, K-FAC) is not valid due to infinite-width limit but taken **on top** of studying the NTK limit. I maybe misreading the reference but indicating large width limit justifies block-diagonal approximation of FIM seems ungrounded.

**Summary Of The Paper:**

The paper empirically studies the effectiveness of Takeuchi’s information criteria (TIC) for deep neural networks near the NTK regime. Since exact computation for TIC (Tr(H^{-1} C)) is often intractable for practical DNNs, the authors explore various approximations of TIC.

The authors’ claimed contributions are:
- Empirical and theoretical evidence that TIC is highly correlated with the generalization gap in DNNs that are close to the NTK regime. Given TIC is not designed for singular models, this is a surprising finding.
- Conduct comprehensive experiments (varying models, datasets, training settings) showing TICs estimate generalization gaps
- Application of these findings to prune poorly performing models in hyperparameter optimization demonstrates this can successfully prevent promising candidates being pruned prematurely.


**Summary Of The Review:**

While study of generalization measure and extensive empirical study is important problem towards understanding generalization behaviour of deep neural networks, the current submission has issues that needs to be addressed before being presented to general audience. 1) The main message / lessons beyond Thomas et al., 2019 is unclear or not existent 2) usefulness of TIC as generalization measure need to explained 3) overall clarity needs to be improved 4) HPO experiments at the currently presented results doesn't demonstrate robustness and requires validation on more diverse settings. The reviewer believes addressing these issues requires major revision and won't be able to  suggest acceptance for ICLR 2022.

---

> ### Author Response · Authors · 2021-11-23
> **Official Response to Reviewer 2vba**
>
> Thank you for taking your time to review our manuscript. Below we address specific comments and questions:
>
> > To estimate TIC, we require a separate data (validation) as well as trained parameter \theta^*for H and C computation. In this case, why not just compute validation acc/loss which we know is a good estimate of generalization performance? [know without training? But needs trained params…]
>
> The generalization gap can be estimated roughly from the training data for a trained model, even without validation data.
> However, in our experiments, the estimation using validation data has a higher estimation performance.
>
> While the validation loss of the validation data has an estimation error of $O(1/\sqrt{n})$, the TIC has the advantage of keeping the estimation error to $O(1/n)$. This is because TIC approaches asymptotically to Cross-Validation, which is usually quite computationally expensive. However, we have achieved a level of speedup that can be used in hyperparameter optimization by using approximate methods.
>
> > Could you provide the argument / reference showing that layer-wise block diagonal structure of Hessian in the NTK regime (section 3.2 part 2)? As far as the reviewer is aware, there’s block diagonal structure for NTK regards to output unit or K-FAC approximation (irrespective of NTK limit) but is not familiar with layer-wise block diagonality.
>
> Layer-wise, or in other words block-diagonal, approximation is valid for NTK regime, as shown in [1].
>
> > The reviewer tried looking for detailed model descriptions used (beyond simple names) for experiments but did not appear in paper nor in code.
>
> We are sorry that this part was incomplete. We have updated the repository.
>
> > Overall, there’s issues with clarity. The paper needs significant proofreading to the point that deters understanding the main message.
>
> Thank you for pointing out it.
> We tried to make the margins of the figures smaller and the text larger in our revision. We also added an explanation of the color bar and a description of Figures 1 and 2.
> We will reconstruct the paper overall to clarify the main point of this study.
>
> > HPO experiments are promising. However, just showing one set of cases in Figure 5, it’s unclear whether the proposed method and described phenomena is robust.
>
> Thank you for pointing out the defect of the experiments.
> We will fix it and add an experiment to properly support the claims of the paper.
>
> > According to remark 2.3., one would consider WAIC/GIC should be a more principled information criteria to be used for models near the NTK regime. TIC is also intractable due to high computational cost. Shouldn’t one also try to approximate the WAIC as similarly be done in the paper? Is there a chance that various approximations taken for TIC make it more like WAIC?
>
> We focus on TIC in this study for two reasons: the definition of GIC is more complicated than TIC, and its calculation is considerably more costly than TIC, and GIC is consistent with TIC when used with the maximum likelihood method.
> As for WAIC, unless we use Bayesian deep learning or something similar, we cannot obtain the posterior distribution, so we must apply a rough approximation such as Laplace approximation.
> We think that both WAIC and TIC are issues that need to be addressed.
>
> [1] Karakida, Ryo, and Kazuki Osawa. "Understanding Approximate Fisher Information for Fast Convergence of Natural Gradient Descent in Wide Neural Networks." Advances in Neural Information Processing Systems 33 (2020).

---

### Official Review · Reviewer_VPJq · 2021-11-03

**Correctness:** 3
**Technical Novelty And Significance:** 2
**Empirical Novelty And Significance:** 2
**Recommendation:** 3
**Confidence:** 4

**Main Review:**

Pros:
-	Empirical study is extensive and refined. Results are obtained from both small and practical scale architectures and image datasets. Approximation using block diagonal matrices is justified by empirical experiments.
-	Minimizing the TIC seems to be a promising guideline for selecting hyperparameters as shown by empirical experiments

Cons:
-	Some previous papers, e.g. Ghorbani et al. (2019) [Sect. 4.1], Gur-Ari et al (2018), Li et al. (2020), show that there is significant alignment in top eigenspace between the Hessian and gradient covariance matrix using various architectures and datasets. Under conjecture that H and C are simultaneously diagonalizable around theta^*, the TIC in Eq. (3) of Prop. 2.1 may be approximated by the sum of product between the eigenvalues of $H^{-1}$ and $C$ matrices. The relative magnitude of the eigenvalues of $H$ and $C$ may be indicative to the generalization gap given that the TIC is a good measure of generalization gap. Does Figure 4 enable one to draw conclusion that the rate of eigenvalues of $H$ and $C$ are constant, so TIC may approximately be a measure of effective dimension?
-	As far as I am concerned, the NTK regime only warrants the convergence to “a” global minimizer, but it does not warrant a unique model minimizer. Therefore, Assumption (A1) may still be unrealistic even in the NTK regime, so (A1) is restrictive. More explanation may be required. In addition, does the solution in “the model has only one possible solution” refer to the empirical minimizer $\hat\theta$ or $\theta^*$?
-	Theory that can be applied on singular models may be more appropriate for deep learning. In particular, recent results show that the singularity, e.g. flatness, of deep neural networks is highly correlated with its generalization performance, and effort has been made to explicitly use this to achieve the state-of-the-art performance, e.g. SAM (https://arxiv.org/abs/2010.01412). As suggested in Remark 2.3, some papers have studied the WBIC of Watanabe by empirically estimating the real log canonical threshold (RLCT) for neural networks, e.g. Murfet et al. (2020). Much effort is still needed to scale up the computation to deep neural networks. RLCT has a fascinating interpretation of the dimension of singularity, which connects with the loss landscape.
As said in Remark 2.1, TIC only applies to non-singular models, so theoretically it is incapable of explaining the generalization of deep learning. However, the empirical results appear to suggest the contrary, at least in NTK regime. Is this just a coincidence, or is there anything that TIC can do beyond that can be covered in theory? Perhaps TIC also affords an interpretation that is related to the loss landscape?
-	Given that many measures have been proposed to explain the generalization gap, e.g. Jiang et al. (2019), the performance of TIC should have been compared with other measures to further document the advantage of TIC.
-	I am unsure how relevant TIC is in selecting hyperparameters, especially when its use is prohibitive on large scale models due to cost, and the validation loss curve of TIC pruning looks similar to that based on the validation loss from panel (b) and (c) of Figure 5. This may not be directly relevant to this paper, but training variance has been a more important issue for generative neural network than for a typical predictive neural network. Therefore, maybe using TIC on hyperparameter tuning for GAN could be a more interesting problem to study.

Minor comments:
-	Typo: 3rd line under Eq. (2), “they a different different …”

**Summary Of The Paper:**

This paper proposes to use TIC as a metric for measuring the generalization gap. Using an extensive set of architectures and datasets, this paper present empirical results that show the correlation between the TIC and generalization gap.

**Summary Of The Review:**

Although this paper presents many empirical results, my main reservation is how much they can afford a deeper theoretical insight that can motivate follow-up studies. In addition, the approximation method to compute the Hessian matrix essentially follows by the existing methods which does not seem very novel.

---

> ### Author Response · Authors · 2021-11-23
> **Official Response to Reviewer VPJq**
>
> Thank you so much for your thoughtful review! Below we address specific concerns:
>
> >  Under conjecture that H and C are simultaneously diagonalizable around theta^*, the TIC in Eq. (3) of Prop. 2.1 may be approximated by the sum of product between the eigenvalues of  $H^{-1}$ and $C$ matrices. The relative magnitude of the eigenvalues of $H$ and  $C$ may be indicative to the generalization gap given that the TIC is a good measure of generalization gap. Does Figure 4 enable one to draw conclusion that the rate of eigenvalues of H and C are constant, so TIC may approximately be a measure of effective dimension?
>
> Thank you for pointing out it.
> First of all, Fig. 4 shows a strong correlation between the TIC estimator and the generalization gap, even in the middle of training.
> Secondly, we agree that it related to the Effective Dimension that the TIC estimator by approximation such as Tr(C)/Tr(H) captures the generalization gap.
> However, we have not thoroughly investigated these relationships yet, so we will leave this as an issue for next time. Thanks for the interesting feedback.
>
> > As far as I am concerned, the NTK regime only warrants the convergence to “a” global minimizer, but it does not warrant a unique model minimizer. Therefore, Assumption (A1) may still be unrealistic even in the NTK regime, so (A1) is restrictive. More explanation may be required. In addition, does the solution in “the model has only one possible solution” refer to the empirical minimizer Θ or Θ ?
>
> You have raised an important question.
> Our description was inadequate in this regard.
> The optimal solution of the model here means that $\hat{\theta}$ is uniquely determined.
> We will reconstruct this proof part to clarify the main point of this study.
>
> > Theory that can be applied on singular models may be more appropriate for deep learning. In particular, recent results show that the singularity, e.g. flatness, of deep neural networks is highly correlated with its generalization performance, and effort has been made to explicitly use this to achieve the state-of-the-art performance, e.g. SAM (https://arxiv.org/abs/2010.01412). As suggested in Remark 2.3, some papers have studied the WBIC of Watanabe by empirically estimating the real log canonical threshold (RLCT) for neural networks, e.g. Murfet et al. (2020). Much effort is still needed to scale up the computation to deep neural networks. RLCT has a fascinating interpretation of the dimension of singularity, which connects with the loss landscape. As said in Remark 2.1, TIC only applies to non-singular models, so theoretically it is incapable of explaining the generalization of deep learning. However, the empirical results appear to suggest the contrary, at least in NTK regime. Is this just a coincidence, or is there anything that TIC can do beyond that can be covered in theory? Perhaps TIC also affords an interpretation that is related to the loss landscape?
>
> Thank you for the helpful suggestion.
> However, as shown in Figs. 13 and 15 of the Appendix, it is difficult to estimate generalization by using Tr(H) and Tr(C) alone.
> The connection with Sharpness and RLCT used in SAM and the interpretation of loss landscape will be the subject of a future paper.
>
> > Given that many measures have been proposed to explain the generalization gap, e.g. Jiang et al. (2019), the performance of TIC should have been compared with other measures to further document the advantage of TIC.
>
> Thank you for pointing out the defect of the experiments.
> We will fix it and add an experiment to properly support the claims of the paper.
>
> > I am unsure how relevant TIC is in selecting hyperparameters, especially when its use is prohibitive on large scale models due to cost, and the validation loss curve of TIC pruning looks similar to that based on the validation loss from panel (b) and (c) of Figure 5.
>
> Within the scope of our experiments, as shown in Table 11 in the Appendix, TIC estimates can be calculated in about 10 seconds for most problem settings (even for cases with a large number of model parameters such as VGG).
> We will experiment with several cases in the HPO problem setting as well. Thank you also for the suggestion of adapting to GAN.

---

### Decision · Program_Chairs · 2022-01-20

**Decision:**

Reject

**Comment:**

While the reviewers acknowledge the broad experimental work done in this paper, they all find several issues, which in their combination show that the paper is simply not in a good enough shape. This impression has not changed during the rebuttal phase and as a result, this is a clear reject.